# Cyclophilin D plays a critical role in the survival of senescent cells

Margherita Protasoni[1,2], Vanessa López-Polo [ID][1], Camille Stephan-Otto Attolini[1], Julian Brandariz [ID][3], Nicolas Herranz[3,4], Joaquin Mateo[3,5], Sergio Ruiz [ID][6], Oscar Fernandez-Capetillo [ID][7,8], Marta Kovatcheva [ID][1,9 ✉] & Manuel Serrano [ID][1,2 ✉]

## Abstract

**Senescent cells play a causative role in many diseases, and their elimination is a promising therapeutic strategy. Here, through a genome-wide CRISPR/Cas9 screen, we identify the gene PPIF, encoding the mitochondrial protein cyclophilin D (CypD), as a novel senolytic target. Cyclophilin D promotes the transient opening of the mitochondrial permeability transition pore (mPTP), which serves as a failsafe mechanism for calcium efflux. We show that senescent cells exhibit a high frequency of transient CypD/mPTP opening events, known as 'flickering'. Inhibition of CypD using genetic or pharmacologic tools, including cyclosporin A, leads to the toxic accumulation of mitochondrial Ca²⁺ and the death of senescent cells. Genetic or pharmacological inhibition of NCLX, another mitochondrial calcium efflux channel, also leads to senolysis, while inhibition of the main Ca²⁺ influx channel, MCU, prevents senolysis induced by CypD inhibition. We conclude that senescent cells are highly vulnerable to elevated mitochondrial Ca²⁺ ions, and that transient CypD/mPTP opening is a critical adaptation mechanism for the survival of senescent cells.**

**Keywords** Cellular Senescence; Mitochondria; Senolytic Therapy; Cyclophilin D; mPTP Flickering
**Subject Category** Organelles

## Introduction

Cellular senescence is a complex response to multiple types of stressors or damage that results in stable cell cycle arrest, even after the stressor is removed and under favourable proliferation conditions (Hernandez-Segura et al, 2018; Gorgoulis et al, 2019). In addition to their inability (or highly reduced capacity) to resume proliferation, senescent cells are characterized by the secretion of a mixture of inflammatory, fibrogenic, and mitogenic soluble factors, known as the senescence-associated secretory phenotype (SASP), together with structural and epigenetic changes, and increased lysosomal biogenesis (Hernandez-Segura et al, 2018; Gorgoulis et al, 2019). In pathological contexts and in advanced aging, the accumulation of senescent cells contributes to chronic inflammation, fibrosis, and reduced regenerative capacity (Daniel Muñoz-Espín, 2021; Chaib et al, 2022; Birch and Gil, 2020). Consequently, the development of novel therapeutics, termed senolytics, capable of clearing senescent cells while sparing other cell populations, has emerged as an appealing strategy for treating senescence-associated disorders. The selective elimination of senescent cells has been shown to delay or prevent numerous age-associated pathologies, including muscular atrophy, cataracts, cancer, and cardiovascular, neurodegenerative, and fibrotic diseases (reviewed in Chaib et al, 2022; Birch and Gil, 2020). Despite their promise, the first generation of senolytics have limitations relating to efficacy, selectivity and toxicity (van Deursen, 2019). The development of improved senolytics requires the identification and understanding of the molecular vulnerabilities of senescent cells.

Emerging evidence suggests that senescent cells acquire highly altered mitochondrial and metabolic adaptations, including changes in morphology and dynamics, impaired respiratory capacity, high production of reactive oxygen species (ROS), and an elevated influx of calcium (Ca²⁺) (reviewed in Protasoni and Serrano, 2023; Martini and Passos, 2023; Ahumada-Castro et al, 2021; Martin et al, 2023). The abnormally elevated influx of Ca²⁺ into the mitochondria of senescent cells occurs mainly through the general pathway that operates in most cells and that involves the endoplasmic reticulum Ca²⁺ efflux channel ITPR2 and the mitochondrial Ca²⁺ influx channel MCU (Ahumada-Castro et al, 2021; Martin et al, 2023; Wiel et al, 2014; Farfariello et al, 2022). Despite these well-known alterations in the mitochondria of senescent cells, the identification of mitochondrial targets for senolysis remains a poorly-explored area.

Through a genome-wide CRISPR-based screening, we have identified *PPIF*, the gene encoding the mitochondrial matrix protein cyclophilin D (CypD) as a novel senolytic candidate. CypD

[1]Institute for Research in Biomedicine (IRB Barcelona), Barcelona Institute of Science and Technology (BIST), 08028 Barcelona, Spain. [2]Cambridge Institute of Science, Altos Labs, Granta Park, Cambridge CB21 6GP, UK. [3]Vall d'Hebron Institute of Oncology (VHIO), Barcelona, Spain. [4]Vall d'Hebron Institute of Research (VHIR), Barcelona, Spain. [5]Vall d'Hebron University Hospital, Barcelona, Spain. [6]Laboratory of Genome Integrity, Center for Cancer Research, National Cancer Institute, NIH, Bethesda, MD 20814, USA. [7]Spanish National Cancer Research Center (CNIO), 28028 Madrid, Spain. [8]Science for Life Laboratory, Division of Genome Biology, Department of Medical Biochemistry and Biophysics, Karolinska Institute, Stockholm, Sweden. [9]IFOM ETS—The AIRC Institute of Molecular Oncology, Milan, Italy. ✉E-mail: marta.kovatcheva@ifom.eu; mserrano@altoslabs.com

is the main positive modulator of the mitochondrial permeability transition pore (mPTP) (Tanveer et al, 1996). The physiological function of the mPTP involves the regulated and transient opening of the pore, also known as flickering (Bernardi and von Stockum, 2012; Hüser and Blatter, 1999; Elrod et al, 2010). Flickering of the mPTP has been observed under normal conditions in neurons and cardiomyocytes, where its main function is to maintain mitochondrial homeostasis by facilitating the rapid exchange of molecules between the mitochondrial matrix and the cytosol (Bernardi and von Stockum, 2012; Hüser and Blatter, 1999). In particular, $Ca^{2+}$ overload poses an important risk for mitochondrial functionality; this risk can be mitigated by the transient opening of the mPTP (Bernardi and von Stockum, 2012; Hüser and Blatter, 1999). The role of the mPTP in cellular senescence, however, has not been previously studied. Here, we report that the survival of senescent cells is highly dependent on CypD-dependent mPTP flickering. We also present mechanistic evidence that mPTP flickering is required to counterbalance the elevated mitochondrial $Ca^{2+}$ influx characteristic of senescent cells. Accordingly, genetic or pharmacologic inhibition of CypD results in toxic accumulation of mitochondrial $Ca^{2+}$ and preferential cell death of senescent cells. This opens novel therapeutic strategies for diseases associated with cellular senescence.

## Results

### Identification of cyclophilin D as a senescence survival gene

As a strategy to identify novel genetic senolytic targets, we employed a previously described Clustered Regularly Interspaced Short Palindromic Repeats/CRISPR Associated Protein 9 (CRISPR/Cas9) system in mouse embryonic stem (ES) cells (Ruiz et al, 2016). Briefly, these cells bear a doxycycline-inducible *Cas9* transgene and were transduced with a lentiviral sgRNA library designed to target 18,424 genes with approximately five independent sgRNAs per gene (90,230 sgRNA in total) (Koike-Yusa et al, 2014). We first devised a system to induce cellular senescence in cultures of differentiated mouse ES cells (mESCs) (Fig. 1A). In particular, mESCs were induced to differentiate into a mixture of cell lineages by the concomitant withdrawal of leukemia inhibitory factor (LIF) and addition of retinoic acid (RA) during 5 days. Differentiation was confirmed by loss of expression of the pluripotency gene *Oct4* (Fig. EV1A) and upregulation of genes representing all three embryonic germ layers, namely ectoderm (*Pax6, Nestin*), mesoderm (*Meox1, Snai1, Fgf5*) and endoderm (*Sox7, Gata4*) (Fig. EV1A). Differentiated-mESCs were then induced to senesce by two independent inducers: the CDK4/6 inhibitor palbociclib (which induces senescence by functionally mimicking the CDK-inhibitory activity of p16) and x-ray irradiation (Fig. 1A). The onset of senescence was confirmed by the expression of the senescence-associated genes *Cdkn2a, Cdkn1a*, and *Il6* (Fig. EV1B), and by the detection of senescence-associated β-galactosidase activity (Fig. EV1C). Finally, we wanted to assess if the resulting population of senescent differentiated-mESCs cells representing the three germ layers was sensitive to two broad spectrum senolytic treatments, navitoclax and dasatinib. Importantly, both navitoclax and dasatinib showed senolytic activity towards palbociclib-senescent

and irradiation-senescent differentiated-mESCs (Fig. EV1C). Therefore, we performed our screening in a mixture of senescent cells representing cell types from all three developmental lineages, with the potential to discover broad spectrum senolytic targets.

Activation of Cas9 was induced by the addition of doxycycline for 10 days in the senescent differentiated-ES cells, as well as in the control differentiated-ES cells not exposed to senescence triggers (Fig. 1A). Genomic DNA was harvested from the different cell populations and the abundance of sgRNA reads before and after doxycycline treatment was compared in each condition (i.e. differentiated, senescent palbociclib, and senescent irradiation. Since all three experimental groups, including differentiated-ES, correspond to predominantly non-dividing cells, we reasoned that those genes whose sgRNA abundance was depleted after doxycycline addition would correspond to genes essential for survival. Moreover, those genes depleted only in senescent cells but not in differentiated cells are candidate senolytic targets. We prioritized genes whose sgRNA abundance was reduced by approximately one-third (i.e. Log2FC ≤ −1.5 when comparing post-doxy vs. pre-doxy); in order to identify broadly applicable senolytic targets, we focused on genes that were depleted in both senescence conditions. Of note, the main subunit of the $Na^+/K^+$-ATPase (ATP1A1), a well-known senolytic target (Guerrero et al, 2019; Triana-Martínez et al, 2019), was among the sgRNAs strongly depleted in the palbociclib-senescent cells (Log2FC ≤ −1.5), although it did not reach this threshold in the irradiation-senescent cells. Only 13 genes had sgRNAs commonly depleted in both palbociclib- and irradiation-senescent cells (Fig. 1B,C). Considering the importance of mitochondria for cellular senescence (Protasoni and Serrano, 2023; Martini and Passos, 2023) and the scarcity of mitochondrial senolytic targets reported to date, we decided to pursue the sole mitochondrial target in the list of candidates. Specifically, we focused on *PPIF*, encoding cyclophilin D (CypD), a mitochondrial peptidyl-prolyl cis-trans isomerase (PPIase) (Fig. 1B,C).

### Senolytic effect of cyclophilin D inhibition

Having identified CypD as a potential senolytic target in differentiated mouse embryonic stem cells, we then moved into human cells of various origins. We induced senescence in normal human lung fibroblasts IMR90 (using X-ray irradiation) and in human lung adenocarcinoma A549 cells (using bleomycin) and analysed them 7 days after the initial exposure. We observed higher steady-state levels of CypD protein in both types of senescent cells compared to their proliferative counterparts (Fig. 2A). To test the senolytic potency of CypD inhibition, we used the previously mentioned irradiated-IMR90 and bleomycin-A549 cells, as well as human melanoma SK-MEL-103 cells (abbreviated as SK-MEL) treated with palbociclib. After induction of senescence for 7 days, senescent cells were transfected with si*PPIF* (a single round of transfection with a commercial mix of 30 siRNAs each targeting a different sequence within the same mRNA and hereafter referred to as siCypD) and transfected cells were analyzed after another period of 7 days. CypD mRNA and protein levels were profoundly reduced (Fig. 2B; Appendix Fig. S1A) and cellular viability decreased by ~40% in all three models of cellular senescence, while proliferative cells remained unaffected despite efficient downregulation of CypD (Fig. 2B; Appendix Fig. S1B). To ensure

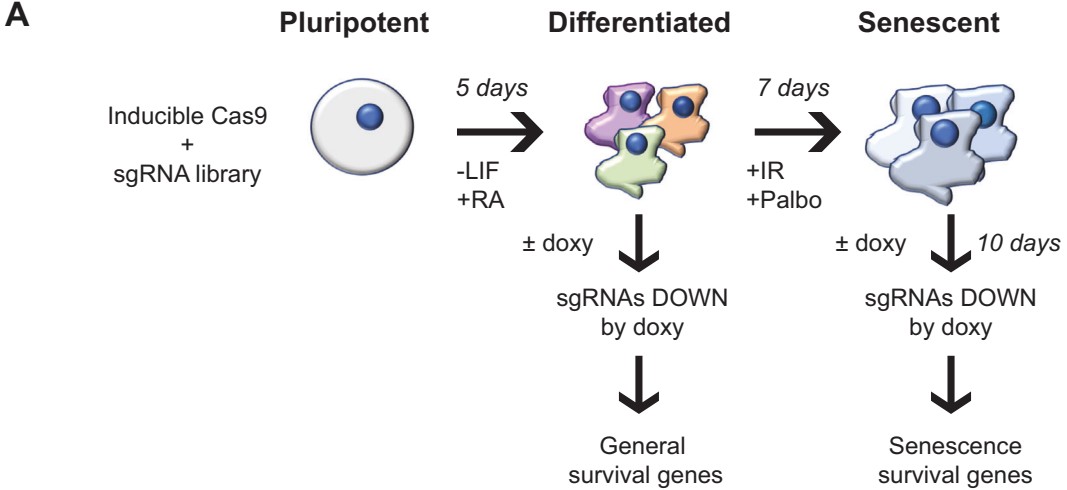

**A**

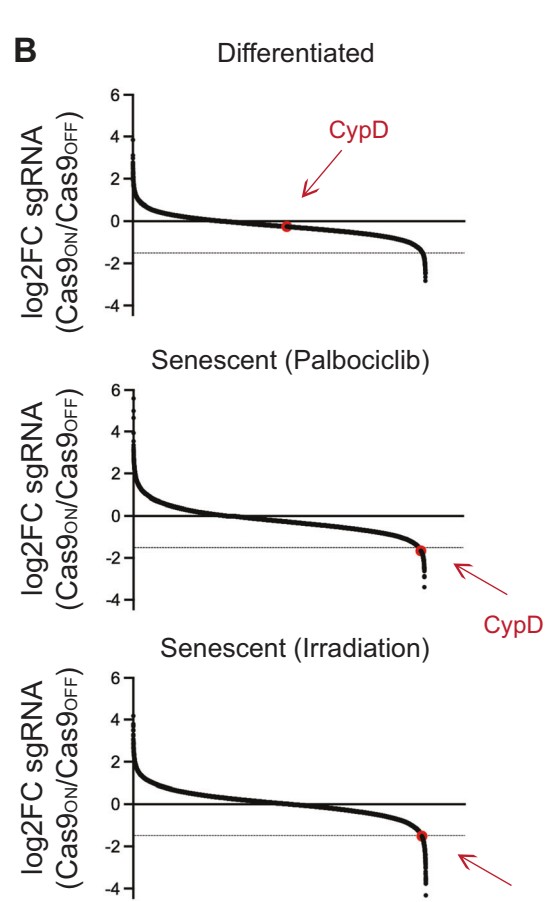

**B**

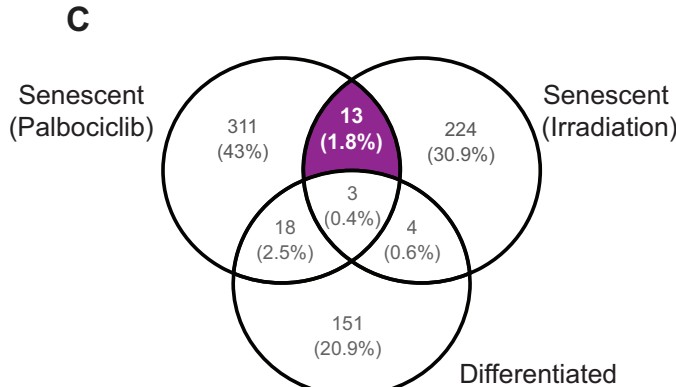

**C**

| Gene | Description | Log2FC (Palbo) | Log2FC (Irradiation) |
|------|-------------|----------------|----------------------|
| *Mcm8* | mini-chromosome maintenance protein | -3.39 | -1.71 |
| *Ufd1l* | degradation of ubiquitinated proteins | -2.46 | -2.78 |
| *Parn* | Poly(A)-specific ribonuclease | -2.45 | -1.76 |
| *Aatf* | Apoptosis-antagonizing transcription factor | -2.45 | -2.30 |
| *Ntsr2* | Neurotensin Receptor | -2.11 | -2.73 |
| *Fam38a* | Mechanosensitive cation channel | -1.87 | -2.26 |
| *Selm* | Selenoprotein | -1.85 | -1.92 |
| *Fam18a* | Intracellular vesicular transport | -1.77 | -1.60 |
| *5031439G07Rik* | Uncharacterised | -1.75 | -2.21 |
| *Trappc3* | Trafficking protein particle complex subunit | -1.71 | -1.91 |
| *Ppif* | mPTP modulator | -1.66 | -1.51 |
| *C4b* | Complement factor | -1.52 | -3.66 |
| *Nudt16* | RNA-binding and decapping enzyme | -1.51 | -2.39 |

**Figure 1. A CRISPR/Cas9 screen in an mESC-based system to identify novel senolytic targets.**

(A) Scheme illustrating the pipeline used for the screening. Pluripotent ES cells transduced with inducible Cas9 and a mouse genome-wide sgRNA library (Ruiz et al, 2016) were induced to differentiate by removing leukemia inhibitory factor (LIF) and adding 0.1 μM retinoic acid (RA) for 5 days. Senescence was then induced using 5 μM palbociclib (Palbo) or 5 Gy irradiation (IR) for 7 days. After this, 2 μg/ml doxycycline (doxy) was added for 10 days to induce Cas9 expression and allow for genomic editing. DNA fragments containing sgRNA sequences were amplified by PCR before and after doxy addition and the relative abundance of each sgRNA was quantified by deep sequencing. The screen was performed in $n = 3$ biological replicates. (B) sgRNA abundance in differentiated cells and cells induced to senescence with Palbociclib or irradiation treatment. The thin gray line indicates the cutoff of -1.5 log2 fold change. The candidate gene PPIF/CypD is indicated in red in each graph. (C) Venn diagram showing genes depleted by a log2 fold change of at least 1.5 in each condition. The 13 genes essential for the survival of senescent cells are defined as those depleted in the cells induced to senescence by palbociclib and irradiation but not in the differentiated condition. The complete list of these genes is reported with the Log2FC value of the respective condition. Source data are available online for this figure.

that the cytotoxic effect of siCypD was specifically senolytic and not generally associated with cell cycle arrest, we induced quiescence in IMR90 and A549 cells by treating cells with sapanisertib (also known as INK128), a selective inhibitor of mTOR, for 7 days (Alhasan et al, 2021). We confirmed that siCypD had no effect on the survival of mTORi-quiescent cells (Fig. 2C). Finally, we assessed the senolytic activity of cyclosporin A (CSA), a well-characterized and commonly used pan-cyclophilin inhibitor, including CypD, which also inhibits the phosphatase calcineurin (PPP3CC) (Amanakis and Murphy, 2020; Patocka et al, 2021). Together with the previously described cellular models, we also tested LNCaP prostate cancer cells treated with abemaciclib, another CDK4/6 inhibitor similar to palbociclib. Treatment of proliferative and senescent (7 days post-trigger) cells with CSA, for 6 additional days, revealed that CSA has senolytic activity in all the four tested human cellular models of senescence, with senolytic indexes (SI, ratio of IC50 values for senescent vs proliferative cells) ranging from 1.59- to 3.21-fold (Fig. 2D). To assess the potential contribution of calcineurin inhibition to the senolytic effect of CSA, we used NIM811, a CSA derivative that lacks inhibitory activity towards calcineurin but retains the ability to inhibit cyclophilins (Wald-meier et al, 2002). Interestingly, NIM811 was also senolytic with SI indexes ranging from 1.87 to 3.75 (Fig. 2E), suggesting that the primary mechanism of senolysis of these drugs is through cyclophilin inhibition.

We then moved to an in vivo model of cellular senescence. It is well established that some chemotherapeutic treatments effectively induce senescence in a subset of cancer cells, and the combined treatment with senolytic drugs results in more efficient tumor regression (Wang et al, 2022). To investigate whether CypD inhibition could eliminate senescent cancer cells in vivo and exhibit anti-tumor activity, we tested the senolytic activity of CSA in a melanoma xenograft model. Xenografts were induced by subcutaneous injection of SK-MEL cells and, upon tumor establishment (when tumors reached 100 mm³, considered day 1), mice were treated with palbociclib (starting at day 1) and/or CSA (starting at day 3) at a dose of CSA previously reported for mouse experiments (Jivrajani et al, 2014). Interestingly, the combination of palbociclib and CSA showed a robust anti-tumor effect, superior to palbociclib alone (Fig. 2F). Notably, treatment with CSA alone did not have any effect on tumor growth, indicating that the positive outcome of CSA treatment depends on the presence of palbociclib-induced senescent cells. Supporting this interpretation, immunohistochemical analysis of the tumors treated with palbociclib showed a substantial increase in the expression of the cell cycle inhibitor p21, a general marker of senescence (Gorgoulis et al, 2019), while tumors from animals treated with both palbociclib and CSA showed significantly

decreased levels of p21 (Fig. 2G). We conclude that, compared to proliferative or quiescent cells, the survival of senescent cells is hypersensitive to the inhibition of CypD.

## Cyclophilin D-dependent mPTP "flickering" in senescent cells

CypD is known to promote the opening of the mitochondrial permeability transition pore (mPTP) (Zhou et al, 2023). The transient opening of the mPTP is known as "flickering" and it is an important mechanism for maintaining mitochondrial health and, thereby, cell fitness (Elrod and Molkentin, 2013). Considering our data implicating elevated CypD in the survival of senescent cells, we wondered if this could be associated with higher levels of mPTP flickering. To visualize mPTP flickering, we recorded mitochondrial membrane potential (ΔΨm) using Tetramethylrhodamine (TMRM) over periods of ~8 min (350 consecutive frames) and we visualized mitochondria using Mitotracker Green (MtG) (Fig. 3A). Flickering is detected by transient losses of TMRM signal that last a few seconds and affect a localized subset of the mitochondrial network. The quantification of transient mPTP openings was performed in proliferative and senescent A549 and IMR90 cells 5 days after a single siScrbl or siCypD transfection. Senescence was induced as previously described (7 days post-trigger) using irradiation for IMR90 and bleomycin for A549 cells. Interestingly, flickering events occurred about threefold more frequently in senescent cells than in proliferative cells, and siCypD strongly reduced the frequency of flickering events (Fig. 3B).

We also monitored mPTP opening using flow cytometry with Calcein-AM and $CoCl_2$ (Yin and Shen, 2022). Calcein-AM penetrates the cytosol and mitochondria of live cells, with $CoCl_2$ selectively quenching the cytosolic signal. During mPTP opening, cobalt enters mitochondria and quenches matrix Calcein-AM fluorescence, allowing us to evaluate mPTP opening. Kinetics analysis showed decreased mitochondrial fluorescence in palbociclib-induced senescent SK-MEL cells after 1 h incubation, while proliferative cells showed no differences (Fig. 3C). As expected, concomitant exposure to CSA blocked mPTP opening and prevented Calcein-AM quenching in senescent cells (Fig. 3C). Therefore, by using two different measurement methods, we conclude that senescent cells present an elevated CypD-dependent mPTP flickering.

## Inhibition of cyclophilin D in senescent cells results in mitochondrial Ca²⁺ accumulation

Previous studies have described mPTP flickering as a $Ca^{2+}$ "emergency exit" during episodes of matrix $Ca^{2+}$ accumulation

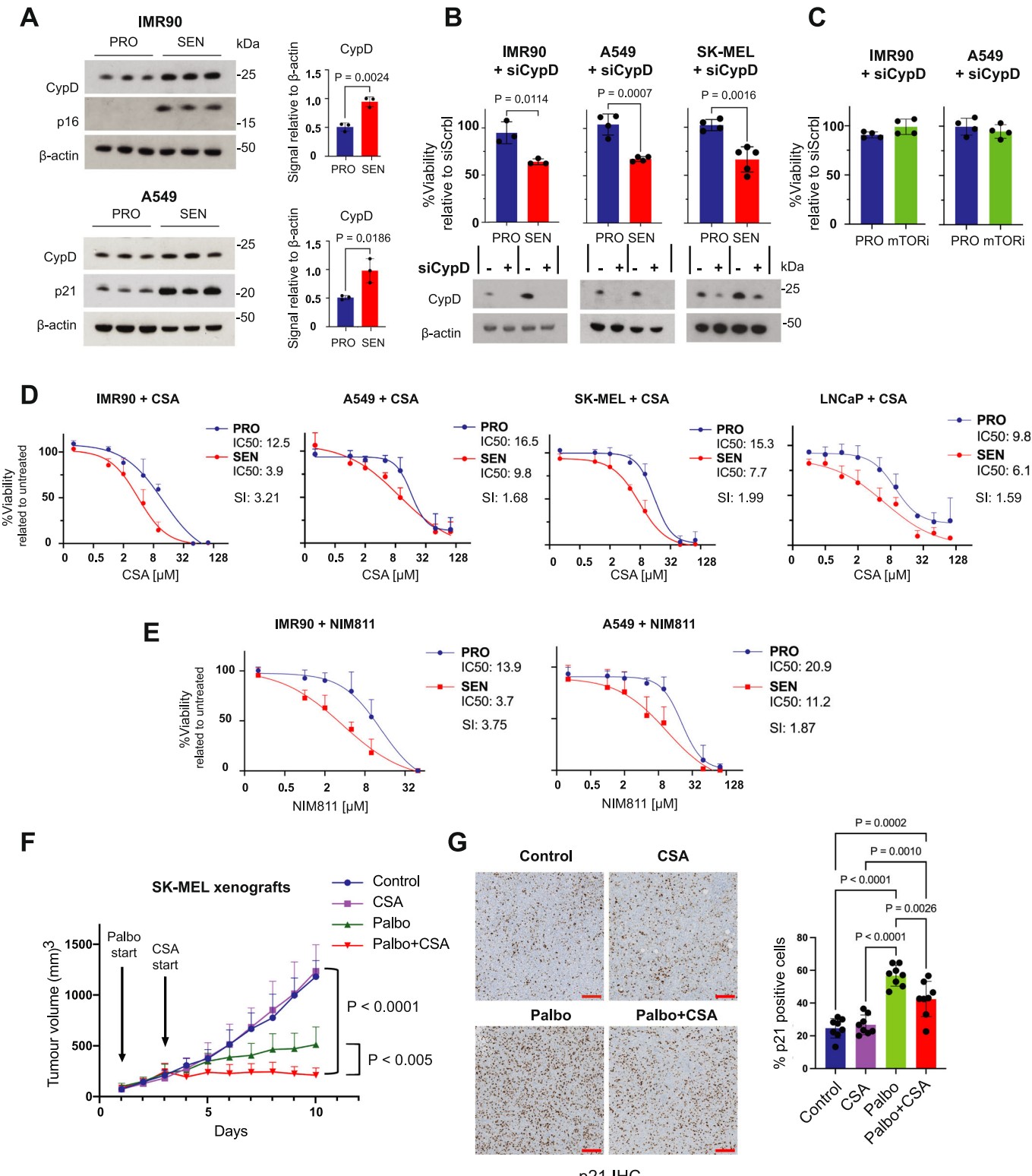

(Wacquier et al, 2019). This failsafe activity could be of particular relevance for senescent cells, given the fact that they constitutively experience an exacerbated mitochondrial $Ca^{2+}$ influx from the endoplasmic reticulum (Ahumada-Castro et al, 2021; Martin et al,

2023; Wiel et al, 2014; Ziegler et al, 2021). Based on this, we hypothesized that CypD-dependent mPTP flickering could be a critical adaptation of senescent cells to allow the release of mitochondrial $Ca^{2+}$ and, thereby, avoid toxic $Ca^{2+}$ accumulation.

◀  **Figure 2.  Depletion or inhibition of CypD specifically eliminates senescent cells in vitro and in vivo.**

(A) Immunodetection of CypD, p16, and p21 by western blots of total cell lysates separated by SDS–PAGE. The graphs show the densitometric quantification of the signals corresponding to each protein normalised to that of the β-actin, from $n = 3$ independent biological replicates of IMR90 and A549 cells. (B) Relative cell viability (%) of proliferating (PRO, blue) or senescent (SEN, red) IMR90, A549, and SK-MEL cells after 7 days of treatment with anti-CypD siRNA, compared to the same cells treated with "scrambled" siRNA (siScrbl). Senescence was induced by treatment with X-ray irradiation (IMR90), or bleomycin (A549), or palbociclib (SK-MEL) for 7 days before the initiation of the treatments with siRNAs. Western blots show the residual levels of CypD in the three cellular cultures on the last day of the experiment. Lanes treated with siCypD are indicated as (+), lanes treated with siScrbl are indicated as (−). $n = 3$–5 independent experiments. (C) Relative cell viability (%) of proliferating (blue) or mTORi-quiescent (green) IMR90 and A549, after 7-days of treatment with anti-CypD siRNA, compared to the same cells treated with siRNA Scrbl. mTORi quiescence was induced with Spanisertib for 7 days before the initiation of the treatments with siRNAs. $n = 4$ independent experiments. (D) Survival curves obtained with increasing concentrations of Cyclosporin A (CSA) in IMR90, A549, SK-MEL, and LNCaP cells, proliferating (PRO) or senescent (SEN), compared to vehicle (DMSO). IC50 values (half maximal inhibitory concentration) and senolytic index (SI: ratio of IC50 values for senescent vs proliferative cells) are indicated for each cell type. Senescence was induced by treatment with ionizing irradiation (IMR90), bleomycin (A549), palbociclib (SK-MEL), or abemaciclib (LNCaP) for 7 days before the initiation of CSA treatment for 6 days. Viability of IMR90, A549, and SK-MEL cells was measured by CellTiter-Glo® Luminescent Cell Viability Assay. Viability of LNCaP cells was measured by Annexin V-BV421 and propidium iodide staining. $n = 3$–4 independent experiments. (E) Survival curves obtained with increasing concentrations of NIM811 in IMR90 and A549, proliferating or senescent, compared to vehicle (DMSO). Senescence was induced by treatment with doxorubicin (IMR90) or bleomycin (A549) for 7 days before the initiation of NIM811 treatment for 6 days. Viability was measured by CellTiter-Glo® Luminescent Cell Viability Assay. $n = 3$–4 independent experiments. (F) Tumor volume ($mm^3$) progression in nude mice over 10 days after oral gavage administration of Palbociclib (from day 1), intraperitoneal injection of CSA (from day 3, or the combination of both (Palbo+CSA), compared to controls. $n = 8$ per group. (G) Representative images of immunohistochemical analysis of p21 and quantification in tumors obtained from the nude mice used in (E). Scale bar = 200 μm. All the graphs show the mean ± SD. Statistical analyses were performed with Unpaired Student's $t$ test in (A–C), and two-way ANOVA multiple comparison with Tukey's correction in (E, F). $P$ values are indicated in the figure. $P$ values in (E) indicate significance from day 7 to end of the experiment. Source data are available online for this figure.

To test this, we analyzed resting mitochondrial $Ca^{2+}$ levels at the individual cell level by fluorescence microscopy using the $Ca^{2+}$-indicator Rhod-2/AM (abbreviated as Rhod-2). To ensure the specificity of this probe and the correct interpretation of the data, we verified that Rhod-2 signal colocalized with Mitotracker Green, and measured mitochondrial membrane potential (ΔΨm), which might influence Rhod-2 fluorescence intensity (Fig. EV2A,B). While bleomycin-induced senescent A549 cells showed lower membrane potential than proliferative cells, ΔΨm was not impacted by CypD downregulation, guaranteeing that direct comparisons between siCypD and siScrbl treated cells are not affected by a different mitochondrial potential. Compared to their proliferative counterparts, CypD-depleted IMR90 and A549 senescent cells showed accumulation of mitochondrial $Ca^{2+}$ 4 days after the initiation of siRNA treatment (Fig. 4A,B). This result was replicated in A549 senescent cells stably expressing the genetic $Ca^{2+}$ sensor CEPIA-2mt (Fig. EV3A). Reinforcing the idea that this phenomenon is specific of senescent cells, $Ca^{2+}$ accumulation was not observed in proliferating cells, nor in mTORi-quiescent cells (Fig. 4C). We conclude that, upon down-regulation of CypD, senescent cells accumulate mitochondrial $Ca^{2+}$, while their non-senescent counterparts are able to maintain normal levels of mitochondrial $Ca^{2+}$.

We also performed $Ca^{2+}$ flux analysis in A549 senescent cells stained with Rhod-2 and challenged them with 10 μM ATP to trigger ER calcium release (Fig. 4D). As expected, Rhod-2 baseline signal was higher in CypD-downregulated senescent cells (4 days post-treatment with siCypD), than in siScrbl-senescent cells. Interestingly, the peak of Rhod-2 signal after ATP stimulation was greatly attenuated in siCypD-senescent cells compared to siScrbl-senescent cells, suggesting that senescent cells with down-regulated CypD may have reached a maximum permitted level of mitochondrial $Ca^{2+}$. The specificity of Rhod-2 in this setup was confirmed by manipulating the main mitochondrial $Ca^{2+}$ trans-porter MCU (mitochondrial calcium uniporter) and one the most important mitochondrial $Ca^{2+}$ efflux channels NCLX (mitochon-drial sodium/calcium antiporter). Treatment of senescent A549 cells with siRNAs targeting MCU or pre-treated with the MCU

inhibitor Ruthenium 360 (Ru360) profoundly reduced the peak of Rhod-2 signal triggered by ATP. Conversely, treatment with the NCLX inhibitor CGP-37157 resulted in sustained high $Ca^{2+}$ levels post-ATP, while the signal in non-treated senescent cells was gradually reduced (Fig. EV3B). These observations confirm that CypD, and also NCLX, are important mechanisms for the extrusion of mitochondrial $Ca^{2+}$ in senescent cells.

To explore whether the elevated $Ca^{2+}$ levels in the mitochondria of senescent cells could be attributed to changes in the steady-state levels of the most important $Ca^{2+}$ transporters, we performed Western blot analyses of several MCU complex subunits (MCU, MICU1, MICU2, and MCUb) and TMBIM5 in proliferative and bleomycin-induced senescent A549. Interestingly, we observed that senescent cells have reduced levels of MCUb and a modest decrease in MICU2 levels, while the levels of the other transporters were similar between senescent and proliferative cells (Fig. EV3C). Considering that MCUb is a dominant-negative subunit of MCU, we speculate that its reduced levels may contribute to the dysregulated $Ca^{2+}$ influx into the mitochondria of senescent cells. We were not able to reliably detect NCLX and LETM1 by immunoblotting, but RT-qPCR analyses did not reveal significant differences between senescent and proliferative cells, with or without downregulation of CypD (Fig. EV3D). The lack of significant differences between cells treated with siCypD or siScrbl in the above-mentioned proteins and mRNAs reinforce the idea that the observed $Ca^{2+}$ alterations are primarily due to mPTP opening inhibition.

As previously reported (Triana-Martínez et al, 2019; Johmura et al, 2021), we found that senescent cells have higher cytosolic $Ca^{2+}$ (measured with Fluo-4) and lower cytosolic pH (measured with BCECF) compared to non-senescent cells (Fig. EV4A–C). Impor-tantly, CypD inhibition did not affect cytosolic $Ca^{2+}$ or pH (Fig. EV4A–C). By performing kinetic analysis of cytosolic $Ca^{2+}$ fluxes using Fura-2, we observed that senescent cells showed a higher peak after ATP stimulus and maintain higher cytosolic $Ca^{2+}$ levels for longer and independently of CypD (Fig. EV4D,E). Together these observations are consistent with the concept that senescent cells have a general accumulation of $Ca^{2+}$, not only in the

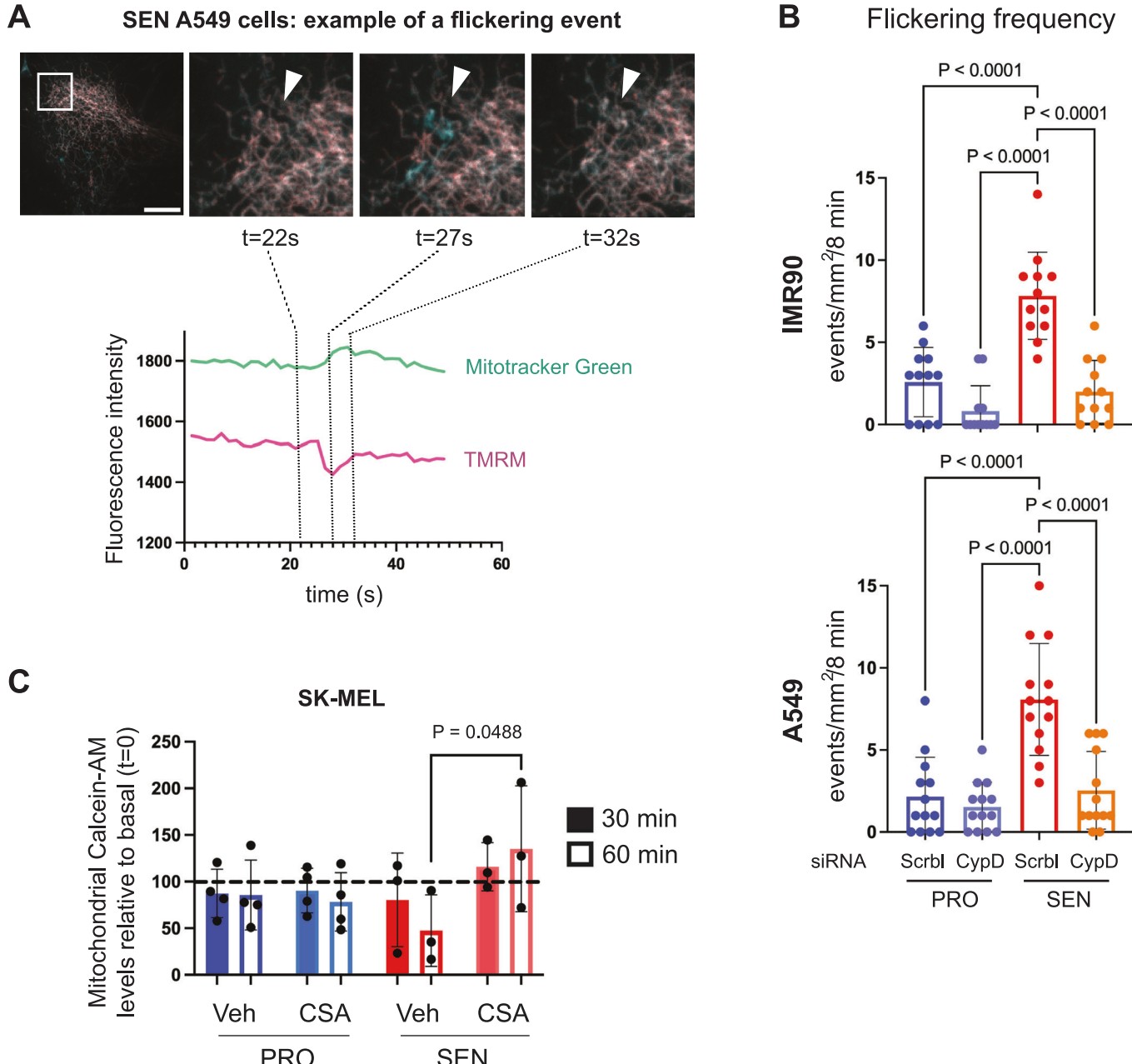

**Figure 3. Transient opening of the mPTP is more frequent in senescent cells than in proliferating cells.**

(A) Representative images of a flickering event in a bleomycin-treated senescent A549 cell. The analysed mitochondria are indicated in the white box in the full image. The white arrowheads show the organelle where the event takes place. TMRM (purple) signal is lost in a spatially and temporally restricted manner during the transient opening of the pore, while it is completely recovered at the end of the event. Mitotracker green signal (cyan) is maintained throughout the event. Scale bar = 50 μm. The graph represents the fluorescence intensity of TMRM and Mitotracker green between 0 and 50 s in the area shown in (A). The observed peak in the TMRM signal is considered as one event in the quantification shown in (B). (B) Quantification of flickering events in proliferating or senescent IMR90 and A549 cells. Senescence was induced with irradiation for IMR90 or bleomycin for A549. Each value represents the number of events observed in one cell during ~8 min (acquisition of 350 images, 0.05 s exposure for the TMRM channel, 0.075 s exposure for the Mitotracker green channel, 5 z-stacks). 10 random areas of 10 × 10 μm were generated and the number of events in each square was calculated. $n = 12$–13 cells per condition, from 3 (proliferating) to 5 (senescent) biologically independent experiments. (C) Flow cytometry assay with Calcein-AM in proliferating (blue) and senescent (red) SK-MEL cells untreated or treated with 20 μM CSA. Senescence was induced with palbociclib. $n = 3$–4 independent experiments. All the values plotted in the graphs are the mean ± SD. Statistical analyses were performed with two-way ANOVA multiple comparison with Tukey's correction. *P* values are indicated in the figure. Source data are available online for this figure.

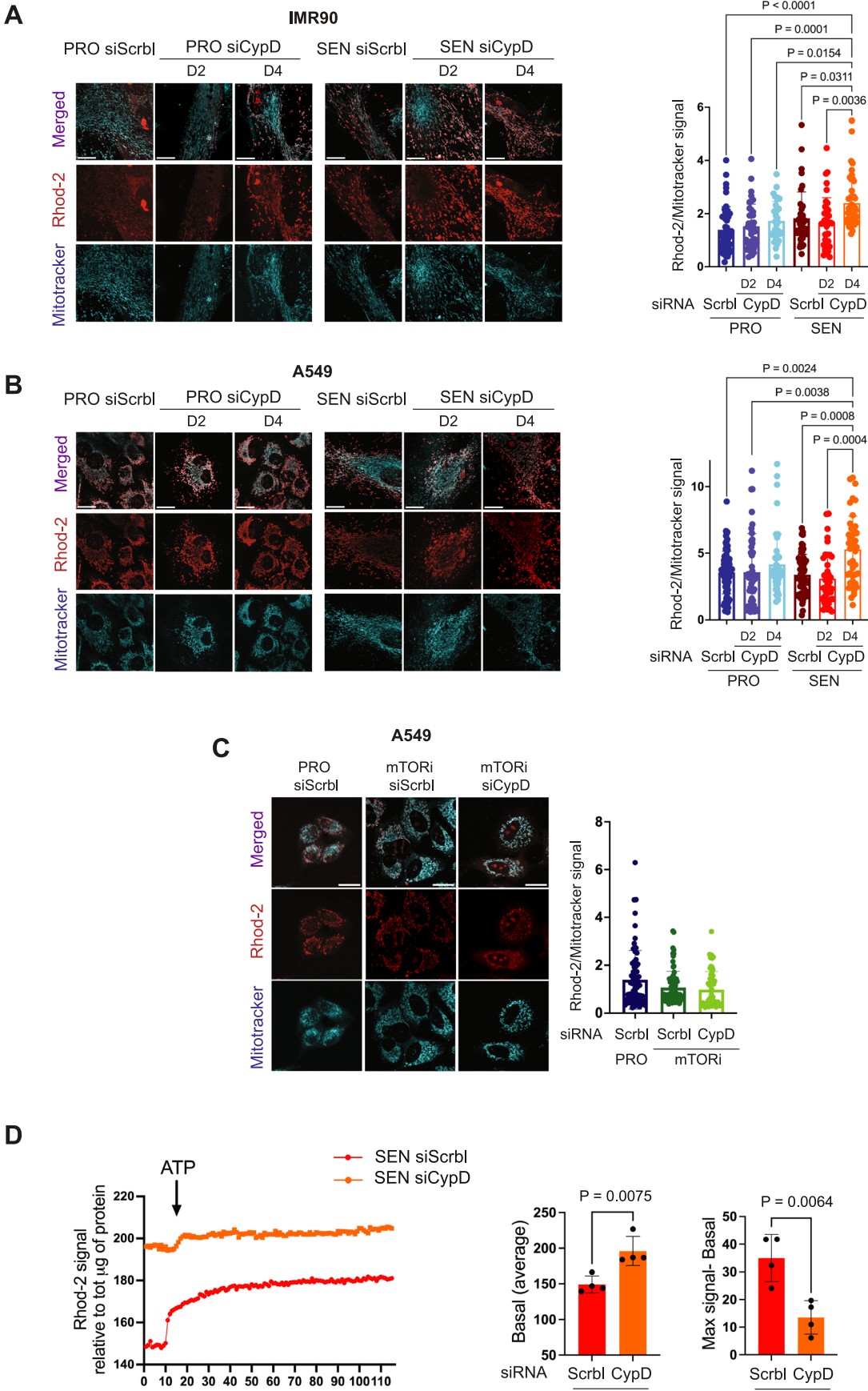

**Figure 4.  Inhibition of mPTP opening leads to mitochondrial calcium accumulation in senescent cells.**

(A) Representative images and quantification of mitochondrial matrix $Ca^{2+}$ levels measured by Rhod-2 in proliferating (blue) and irradiation-induced senescent (red) IMR90 cells, 2 and 4 days after treatment with siRNA against CypD or siRNA Scrbl. Live-cell images were acquired of cells simultaneously stained with Rhod-2 and Mitotracker green. $n = 36–50$ cells from a total of 4 independent experiments. Rhod-2 signal was normalised to mitochondrial mass (Mitotracker green signal). (B) Representative images and quantification of mitochondrial matrix $Ca^{2+}$ levels in proliferating (blue) and bleomycin-induced senescent (red) A549 cells, 2 and 4 days after treatment with siCypD or siScrbl. Live-cell images were acquired of cells simultaneously stained with Rhod-2 and Mitotracker green. $n = 39–91$ cells from a total of 4 independent experiments. (C) Representative images and quantification of mitochondrial matrix levels in proliferating (blue) and mTORi-quiescent (green) A549 cells treated with siCypD or siScrbl for 6 days. Live-cell images were acquired of cells simultaneously stained with Rhod-2 and Mitotracker green. $n = 57–67$ cells from a total of four independent experiments. (D) Average Rhod-2 fluorescent trace over time, average baseline levels, and difference between maximal fluorescent signal and baseline in A549 senescent cells expressing (red trace) or depleted of CypD (orange trace) before and after 10 µM ATP stimulus. $n = 4$ independent experiments. Cells depleted of CypD had been transfected with siCypD for 5 days. All the values plotted in the graphs are the mean ± SD. Statistical analyses were performed with two-way ANOVA multiple comparison with Tukey's correction. *P* values are indicated in the figure. Scale bars = 20 µm. Source data are available online for this figure.

mitochondria but also in the cytosol and in the ER, but inhibition of CypD selectively affects the levels of mitochondrial $Ca^{2+}$.

## Cyclophilin D inhibition affects mitochondrial morphology in senescent cells

Alterations in mitochondrial $Ca^{2+}$ can lead to mitochondrial dysfunctions and remodeling (Brookes et al, 2004). In particular, elevated mitochondrial $Ca^{2+}$ triggers mitochondrial fragmentation (Hom et al, 2007, 2010; Chakrabarti et al, 2017) and swelling (Halestrap et al, 1986). Based on this, we decided to further characterize the effect of CypD depletion on mitochondrial structure and functionality in senescent cancer cells and fibroblasts after 5 days of CypD downregulation by siRNA treatment, a time point at which matrix $Ca^{2+}$ accumulation can be detected, but senolysis is still low. We observed that siCypD significantly changed mitochondrial morphology in senescent cells, but not in proliferative cells. Specifically, mitochondrial volume and length were significantly reduced by siCypD in senescent cells (Fig. 5A–D), which are suggestive of mitochondrial fragmentation. Therefore, CypD inhibition in senescent cells recapitulates morphological perturbations characteristic of $Ca^{2+}$ overload (Hom et al, 2007, 2010; Chakrabarti et al, 2017), further substantiating the idea that CypD contributes to senescent cell survival through its role in buffering mitochondrial $Ca^{2+}$ levels.

## Mitochondrial respiration, ROS and protein folding upon Cyclophilin D inhibition

Considering that the transient opening of the mPTP locally dissipates the mitochondrial $H^+$ gradient and that matrix $Ca^{2+}$ can affect cellular metabolism, we wondered if the inhibition of CypD in senescent cells would have a detectable impact on the global function of the electron transport chain. We evaluated mitochondrial respiration by measuring oxygen consumption rate in proliferative and senescent cells (IMR90 and A549) under the same conditions described in our previous experiments. Although maximal and spare respiration were higher in senescent cells compared to proliferative cells, the inhibition of CypD did not affect these values (Fig. 5E).

Reactive oxygen species (ROS) levels are considerably higher in senescent cells compared to proliferative cells (Correia-Melo and Passos, 2015) and ROS are a main activator of mPTP opening (Zorov et al, 2014). We wondered if inhibition of CypD in senescent cells could affect mitochondrial ROS levels (measured

using MitoSox). We found that mitochondrial ROS levels in senescent cells are not significantly affected by CypD depletion (Fig. 5F).

In addition to its function in regulating the mPTP, another important function of CypD is to facilitate the folding of mitochondrial proteins (Andreeva et al, 1999; Callegari and Dennerlein, 2018). In this regard, inhibition of CypD may promote the accumulation of unfolded proteins in the mitochondria and the activation of the mitochondrial unfolded protein response ($UPR^{mt}$) (Andreeva et al, 1999; Callegari and Dennerlein, 2018). To assess the mitochondrial chaperone function of CypD, we measured the mRNA levels of markers known to be transcriptionally upregulated during $UPR^{mt}$ (Seiferling et al, 2016). We observed that none of the markers tested (*HSP60, DNAJA3, CLPP, TXN2, mtHSP70, LONP1*) were significantly affected by siCypD in proliferative or senescent IMR90 and A549 cells (Fig. EV5A,B). Collectively, these data indicate that the effects of CypD reduction in senescent cells are largely due to its effects on mitochondrial calcium levels, and are not reflected on other known functionalities of CypD.

## Cyclophilin D inhibition in senescent cells leads to pyroptosis

We wondered about the mechanism of cell death elicited by CypD inhibition in senescent cells. Mitochondrial destabilization by $Ca^{2+}$ overload is a paradigmatic trigger of the NLPR3 inflammasome and caspase 1-dependent cell death, also known as pyroptosis (Horng, 2014; Swanson et al, 2019). Based on our above observations, we hypothesized that the depletion of CypD would trigger caspase-1-dependent cell death. To test this possibility, we incubated CypD-depleted proliferative and senescent IMR90 and A549 cells (generated in the same manner as in previous experiments) with different concentrations of a caspase-1 and -4 inhibitor (VX-765) known to prevent pyroptosis (Zahid et al, 2019). We observed that the senolytic effect of CypD depletion was significantly inhibited by VX-765 in a concentration-dependent manner, suggesting that the pyroptotic cell death program plays a role in CypD-mediated senolysis (Fig. 5G).

## Inhibition of NCLX recapitulates the senolytic effect of Cyclophilin D inhibition

We next sought to further explore the impact of $Ca^{2+}$ accumulation on the viability of senescent cells. We employed two opposing pharmacological approaches; namely, reducing $Ca^{2+}$ entry into

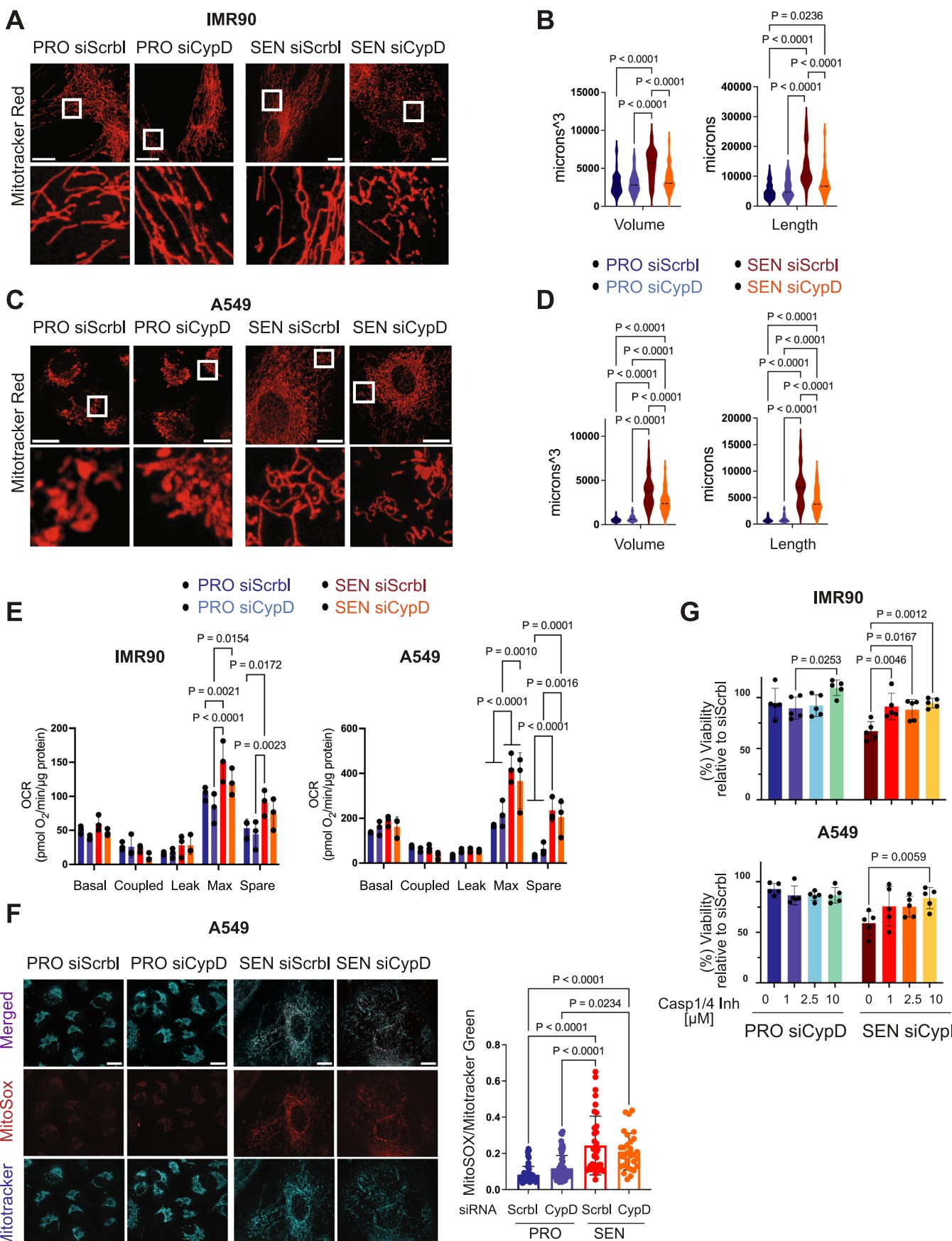

**Figure 5.  Cyclophilin D depletion induces mitochondrial morphology alterations and cell death by pyroptosis but does not significantly impact ETC capacity and ROS production.**

(A) Representative images of mitochondrial morphology in proliferating and senescent IMR90 cells after treatment with siRNA Scrbl or siRNA against CypD. Mitochondria were labelled with Mitotracker red. The white boxes indicate the zoomed-in areas. (B) Violin plots of total mitochondrial volume and total mitochondrial length from cells represented in (A). $n = 39$–49 cells from a total of three independent experiments. (C) Representative images of mitochondrial morphology in proliferating and senescent A549 cells after treatment with siScrbl or siCypD. Mitochondria were labelled with Mitotracker red. The white boxes indicate the zoomed-in areas. (D) Violin plots of total mitochondrial volume and total mitochondrial length from cells represented in (C). $n = 50$–60 cells from a total of 3–4 independent experiments. (E) Oxygen consumption rate (OCR) profiles of proliferating (blue) and senescent (red) IMR90 and A549 cells treated with siScrbl or with siCypD for 5 days. Basal respiration (Basal), ATP-linked respiration (Coupled), leak, maximal respiration (Max), spare respiratory capacity (Spare) are indicated. $n = 3$ biological replicates. Values were normalised to protein amount. (F) Representative images and quantification of mitochondrial ROS levels in proliferating and senescent A549 cells treated with siScrbl or with siCypD for 5 days. Cells were stained with MitoSOX and MitoTracker green and imaged live. $n = 30$–63 cells from a total of 3–4 independent experiments. (G) Relative cell viability (%) of proliferating (blue) or senescent (red) A549 and IMR90 after 7-days of treatment with siCypD, alone or in combination with 1, 2.5 or 10 µM Caspase 1/4 inhibitor (VX-765), compared to the same cells treated with siScrbl. $n = 5$ biological replicates. All the values plotted in the graphs E-G are the mean ± SD. Statistical analyses were performed with two-way ANOVA multiple comparison with Tukey's correction. $P$ values are indicated in the figure. Scale bars = 20 µm. Source data are available online for this figure.

mitochondria by using an inhibitor of the main mitochondrial $Ca^{2+}$ transporter MCU (Woods and Wilson, 2020); and blocking the NCLX transporter, which is the primary efflux pathway for mitochondrial $Ca^{2+}$ (Griffiths, 2009; Wei et al, 2012) (Fig. 6A).

Previous investigators have identified MCU as critical for the establishment of senescence (Woods and Wilson, 2020). We wondered if inhibiting MCU with Ruthenium 360 (Ru360) (de Jesús García-Rivas et al, 2005; Paillard et al, 2018) would prevent cell death induced by siCypD in senescent cells (Fig. 6A). Ru360 has been extensively reported in the literature to significantly reduce mitochondrial $Ca^{2+}$ uptake and basal levels (de Jesús García-Rivas et al, 2005; Paillard et al, 2018; Wei et al, 2012). Senescent cells (7 days after treatment with the senescence trigger) were exposed to Ru360 for 4 additional days before measuring mitochondrial $Ca^{2+}$ levels using Rhod-2. The culture medium containing the inhibitor was changed every 48 h. Consistent with previous studies, Ru360 significantly reduced basal $Ca^{2+}$ levels in senescent A549 cells and substantially rescued $Ca^{2+}$ accumulation in CypD-depleted senescent cells (Fig. 6B). Moreover, senescent A549 and SK-MEL cells transfected with siCypD were concomitantly treated with Ru360 for 6 days and this partially rescued the viability of CypD-depleted senescent cells (Fig. 6C).

Finally, to mimic the $Ca^{2+}$ accumulation caused by CypD inhibition (see above), cells were treated with an inhibitor of NCLX, namely CGP-37157 (abbreviated CGP) (Griffiths, 2009) (Fig. 6A). Exposure of senescent A549 cells to CGP for 4 days caused an increase in mitochondrial $Ca^{2+}$ comparable to the one induced by CSA (Fig. 6B). Importantly, senescent A549 and IMR90 cells treated with CGP for 6 days exhibited similar loss of viability as cells treated with siCypD for 7 days (Fig. 6C). Furthermore, the combination of CSA and CGP showed stronger senolytic effect compared to each treatment alone (Fig. 6D). To further substantiate that CSA and CGP share a common mechanism of senescence-specific toxicity, we asked if CGP produced the same morphological changes in the mitochondria of senescent cells as those observed upon CypD inhibition. Interestingly, treatment of senescent IMR90 and A549 cells with CGP also produced a reduction in mitochondrial volume and length, while the mitochondria of proliferative cells remained unaffected (Fig. 6E,F). To confirm and reinforce these observations, we genetically downregulated NCLX and MCU by transfecting senescent and proliferative cells with specific siRNA for 7 days. Viability assays indicated that siNCLX produced a loss of survival in senescent cells, but not in proliferative cells (Fig. 6G). In the case of senescent A549 cells, the loss of viability by siNCLX was indistinguishable from that elicited by siCypD. Also, siMCU partially

rescued the loss of viability induced by siCypD (Fig. 6G). Morphological analyses of senescent mitochondria treated with siNCLX recapitulated the results obtained with pharmacological inhibition of NCLX or with siCypD, that is a reduction in mitochondrial volume and length (Fig. 6H–J). Together, these results strongly reinforce the concept that senescent cells have an abnormal influx of mitochondrial $Ca^{2+}$ that requires counterbalance by efflux through CypD/mPTP and NCLX. As a consequence of this vulnerability, inhibition of mPTP and NCLX, alone or combined, is toxic for senescent cells.

# Discussion

Senescent cells can be regarded as cells that have adapted to survive in the face of high levels of stress and damage. Understanding these adaptations is important to identify senescence-specific vulnerabilities and, thereby, pharmacologic strategies to eliminate them. In our study, we conducted a CRISPR/Cas9-based screening in murine embryonic stem cells (mESCs) to uncover potential novel senolytic targets, and we focused on the top mitochondrial target that we found, namely, cyclophilin D (CypD). Our rationale for pursuing a mitochondrial target was that the mitochondrial compartment of senescent cells is critical for the establishment and maintenance of the senescence phenotype (Martini and Passos, 2023), however, the molecular adaptations of mitochondria in senescent cells remain poorly understood. Here, we characterize the critical role of CypD in the mitochondria of senescent cells and identify actionable targets with therapeutic potential.

CypD is currently recognized as the main regulator of the mPTP, inducing pore opening by sensitizing it to $Ca^{2+}$, inorganic phosphate, and/or ROS (Amanakis and Murphy, 2020). Transient mPTP opening or 'flickering' allows the exit of $Ca^{2+}$, ROS, and other potentially toxic molecules from the mitochondrial matrix into the cytosol, thereby preventing the overload of molecules that could compromise mitochondrial function (Bernardi and von Stockum, 2012). Indeed, mPTP flickering is relevant for cellular homeostasis, signaling, and cell fate (Bernardi et al, 2023; Ying et al, 2018; Hom et al, 2011). Interestingly, we report by direct observation of local mitochondria depolarization that the events of transient mPTP opening are significantly more frequent in senescent cells than in proliferative cells and are abrogated by CypD inhibition. More importantly, senescent cells are more vulnerable than non-senescent cells (proliferative or quiescent) to cell death induced by CypD inhibition, either genetically using siRNA or pharmacologically using cyclosporin A (CSA) or

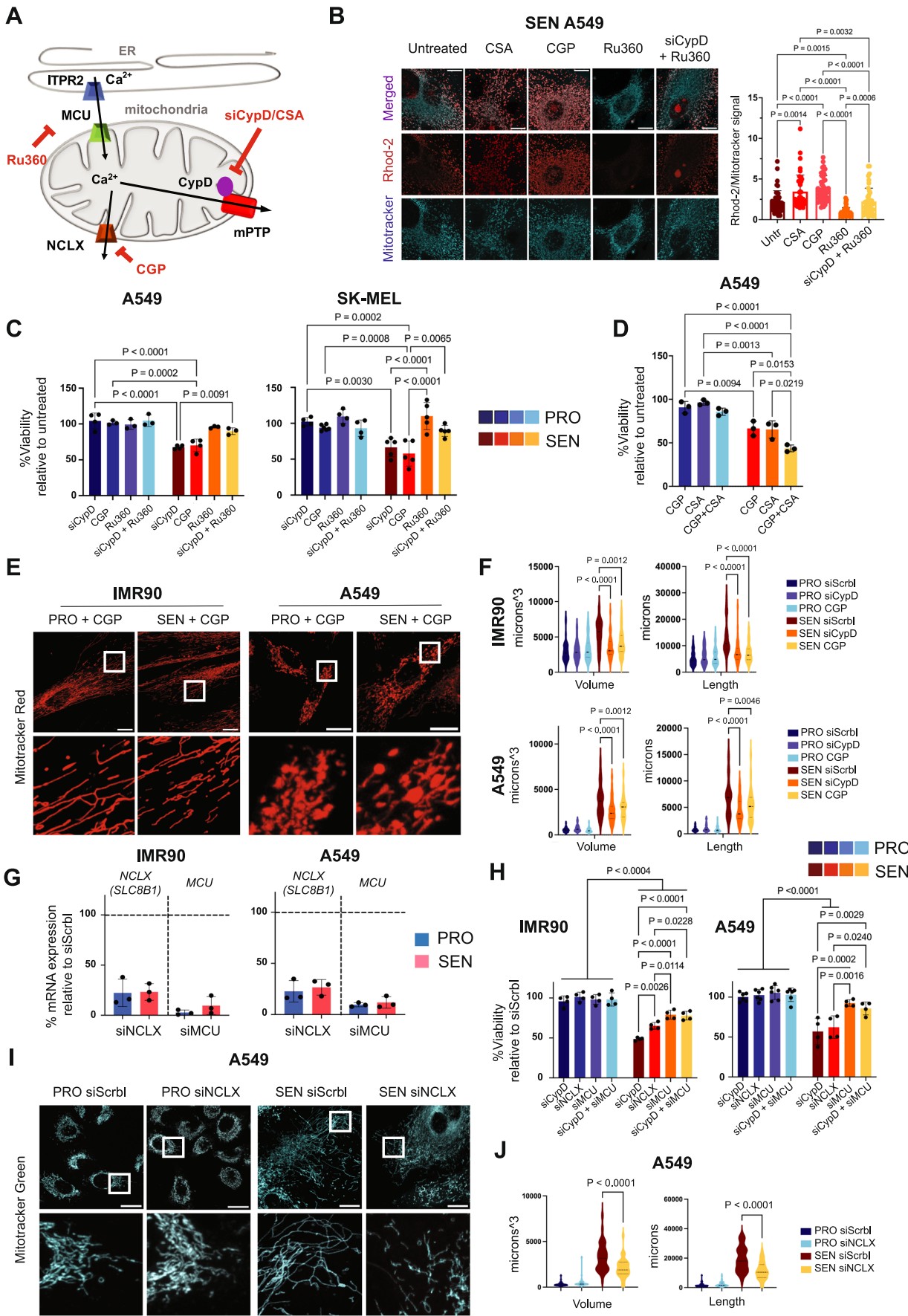

◀ **Figure 6. Modulation of mitochondrial calcium levels can mimic or rescue the phenotype observed in cyclophilin D-depleted cells.**

(A) Schematic representation of $Ca^{2+}$ fluxes in the mitochondrial matrix. $Ca^{2+}$ is stored in the ER and released through the inositol 1,4,5-triphosphate receptors (ITPR2) at the ER-mitochondria contact sites. Here, the mitochondrial calcium uniporter complex (MCU) in the IMM uptakes $Ca^{2+}$ into the matrix. $Ca^{2+}$ exits the matrix mainly through the mitochondrial $Na^+/Ca^{2+}/Li^+$ exchanger (NCLX) and, in case of overload, can be released by the transient opening of the mitochondrial permeability transition pore (mPTP), activated by CypD. CGP-37157 pharmacologically inhibits NCLX, causing an increase in resting matrix $Ca^{2+}$, while Ru360 blocks the activity of MCU, reducing the entry and the accumulation of $Ca^{2+}$ in the organelle. (B) Representative images and quantification of mitochondrial matrix $Ca^{2+}$ levels in senescent A549 cells after 5 days of treatment with CGP-37157, CSA, Ru360, siRNA against CypD in combination with Ru360, or untreated as indicated. Live-cell images were acquired of cells simultaneously stained with Rhod-2 and Mitotracker green ($n = 43$–$47$ cells from a total of four independent experiments). Rhod-2 signal was normalised to mitochondrial mass (mitotracker green signal). The values plotted in the graphs are the mean ± SD. Scale bar = 20 μm. (C) Relative cell viability (%) of proliferating (blue) or senescent (red) A549, and SK-MEL cells after treatment with anti-CypD siRNA (7 days), 10 μM CGP-37157 or 10 μM Ru360 (6 days), or anti-CypD siRNA in combination with Ru360, compared to the same cells treated with siRNA Scrbl. $n = 3$–$5$ independent experiments. The values plotted in the graphs are the mean ± SD. (D) Relative cell viability (%) of proliferating (blue) or senescent (red) A549 after 6 days of treatment with 10 μM CGP-37157, 5 μM CSA, or both, compared to untreated cells. $n = 3$ independent experiments. The values plotted in the graphs are the mean ± SD. (E) Representative images of mitochondrial morphology in proliferating and senescent IMR90 and A549 cells after treatment with 10 μM CGP-37157. Mitochondria were labelled with Mitotracker red. Scale bar = 20 μm. (F) Violin plots of total mitochondrial volume and total mitochondrial length of cells represented in (E). Data from both proliferating and senescent cells treated with only siRNA Scrbl or anti-CypD siRNA were taken from the experiment shown in Fig. 5A–D and used as a basis for comparison. $n = 51$ (IMR90) or $58$–$63$ (A549) cells from a total of three independent experiments. (G) Relative mRNA expression of NCLX and MCU in proliferating (blue) or senescent (red) IMR90 (A) and A549 (B) cells, treated with anti-NCLX or anti-MCU siRNA for 7 days. Residual mRNA is shown as percentage of siRNA Scrbl. Signals were normalised to that of β-actin. $n = 3$ biologically independent samples. (H) Relative cell viability (%) of proliferating (blue) or senescent (red) IMR90, and A549 cells after 7 days treatment with anti-CypD, anti-NCLX, or anti-MCU siRNA, or anti-CypD and anti-MCU siRNA in combination, compared to the same cells treated with siRNA Scrbl. $n = 4$–$6$ independent experiments. (I) Representative images of mitochondrial morphology in proliferating and senescent A549 cells after treatment with siRNA against NCLX. Mitochondria were labelled with Mitotracker green. Scale bar = 20 μm. (J) Violin plots of total mitochondrial volume and total mitochondrial length of cells represented in (I). $n = 35$–$60$ cells from a total of three independent experiments. The values plotted in the graphs are the mean ± SD. Statistical analyses were performed with two-way ANOVA multiple comparison with Tukey's correction. $P$ values are indicated in the figure. Source data are available online for this figure.

NIM811 (a CSA-related compound that lacks inhibition of calcineurin (Waldmeier et al, 2002)). We also present evidence for the senolytic activity of CypD inhibition using CSA in an in vivo model of cellular senescence. While in several pathological models of acute damage the inhibition of mPTP opening has cytoprotective effects due to its role in reducing necrotic cell death, such as in cardiac and cerebral ischemia/reperfusion (Zhang et al, 2019; Matsumoto et al, 1999; Schinzel et al, 2005), our data present a novel scenario in which sustained inhibition of flickering is detrimental specifically for senescent cells.

It has been reported that the mitochondria of senescent cells are subject to an abnormally high influx of $Ca^{2+}$ from the endoplasmic reticulum (Wiel et al, 2014; Ziegler et al, 2021). Based on this, we hypothesized that mPTP flickering might play a protective role against mitochondrial $Ca^{2+}$ overload. Indeed, we observed a progressive accumulation of mitochondrial $Ca^{2+}$ levels in senescent cells after CypD inhibition. Mitochondrial $Ca^{2+}$ overload is well-known to trigger mitochondrial destabilization leading to pyroptosis (Horng, 2014; Swanson et al, 2019). Accordingly, when we treated cells with a caspase 1/4 inhibitor, the viability of CypD-depleted senescent cells was completely rescued, indicating that pyroptosis plays an important role in this type of senolysis. Notably, although ROS is another important trigger of pyroptosis (Horng, 2014; Swanson et al, 2019) and ROS levels are constitutively high in senescent cells, depletion of CypD did not further increased ROS levels in senescent cells. This reinforces the concept that senolysis mediated by CypD depletion is mediated by mitochondrial $Ca^{2+}$ accumulation.

We also observed that CypD depletion induced remodeling of the mitochondrial network in senescent cells. Particularly, the organelles became significantly shorter in both cancer cells and fibroblasts. Notably, mitochondrial fragmentation is a hallmark of $Ca^{2+}$ overload (Hom et al, 2007, 2010; Chakrabarti et al, 2017). To gain further insight into the toxic effects of increased matrix $Ca^{2+}$ levels in different cell populations, we used specific inhibitors and genetic approaches to modulate mitochondrial $Ca^{2+}$ transport. We treated cells with Ru360, a potent inhibitor of $Ca^{2+}$ uptake through the MCU complex, which effectively reduced mitochondrial $Ca^{2+}$ levels and rescued senescent

cancer cells from the detrimental effects of CypD downregulation. On the contrary, CGP-37157, a selective inhibitor of the mitochondrial $Na^+/Ca^{2+}$ exchanger NCLX, resulted in a phenotype similar to that observed in CypD-depleted cells in terms of $Ca^{2+}$ accumulation, cell viability, and alterations in mitochondrial morphology. Genetic downregulation of MCU and NCLX by siRNA transfection recapitulated these results. We interpret that compensation of the elevated influx of $Ca^{2+}$ in the mitochondrial matrix of senescent cells requires both $Ca^{2+}$ efflux mechanisms, NCLX and mPTP. Overall, these findings provide additional evidence for the involvement of mitochondrial $Ca^{2+}$ dynamics in the vulnerability associated with CypD depletion in senescent cells and for the potential of this pathway as a senolytic target.

Cyclosporin A (CSA) is a pan-cyclophilin inhibitor that blocks the CypD peptidyl-prolyl-isomerase active site and interferes with the interactions of CypD with the inner mitochondrial membrane and with other proteins (Schiene-Fischer et al, 2022). CSA is commonly used as immunosuppressant in current medical practice, but it presents important toxicities, particularly after long periods of administration (Patocka et al, 2021; Wu et al, 2018). In addition to being a pan-cyclophilin inhibitor, CSA is a molecular glue that associates cyclophilin A with calcineurin, thereby also inhibiting calcineurin which is key for the immunosuppressive activity of CSA (Geiger et al, 2022). Here, we have used NIM118, a CSA-related compound that lacks inhibition of calcineurin (Waldmeier et al, 2002) and we have observed that it possesses comparable, or even stronger, senolytic activity compared to CSA. It is conceivable that a specific CypD inhibitor will have lower toxicities and more efficacy. In this regard, mitochondrially-targeted CypD inhibitors have been recently developed (Zhang et al, 2019; Ikeda et al, 2016) and could serve as promising pharmacological tools for this task.

In conclusion, our findings shed light on the intricate relationship between CypD, transient mPTP opening, and cellular vulnerabilities in senescence. They also offer a new candidate for senolytic therapeutic interventions, CypD, with consistent and comparable effects in different models. Further investigation into the mechanisms and implications of flickering mPTP could lead to the development of innovative therapeutic interventions for senescence-related disorders.

# Methods

## Reagents and tools table

| Reagent/resource | Reference or source | Identifier or catalog number |
|---|---|---|
| **Experimental models** | | |
| A549 | ATCC | |
| IMR90 | ATCC | |
| SK-MEL-103 | Memorial Sloan Kettering Cancer Center | |
| LNCaP | ATCC | |
| Athymic Nude Mouse | Charles River France | |
| **Recombinant DNA** | | |
| pCMV-CEPIA-2mt | Masamitsu Iino Lab | Addgene |
| **Antibodies** | | |
| Anti-CypD | Santa Cruz | sc-376061 |
| Anti-P21 | Santa Cruz | sc-397 |
| Anti-P16 | Callbiochem | NA29 |
| Anti-β-actin | Abcam | ab8227 |
| Anti-p21WAF1/Cip1 | Dako-Agilent | M7202 |
| Anti-IgG1 | Abcam | ab18443 |
| Anti-MCU | Cell Signaling | 14997S |
| Anti-MCUb | Atlas Antibodies | HPA048776 |
| Anti-MICU1 | Abcam | ab224161 |
| Anti-MICU2 | Abcam | ab101465 |
| Anti-TMBIM5 | Proteintech | 16296-1-AP |
| Anti-GAPDH | Thermofisher | G9545 |
| Anti-mouse (WB) | Agilent Technologies | P044701-2 |
| Anti-rabbit (WB) | Agilent Technologies | P044801-2 |
| Anti-mouse (IHC) | Abcam | ab133469 |
| **Oligonucleotides and other sequence-based reagents** | | |
| | FWD | REV |
| CypD | GAAGGCAGATGTCGTCCCAAA | GGAAAGCGGCTTCCGTAGAT |
| HSP60 | TGCCAATGCTCACCGTAAG | ACTGCCACAACCTGAAGAC |
| DNAJA3 | TTTGGCGAGTTCTCATCCTCT | TTGCAGCTTGATTGAATGTCAAC |
| CLPP | GCAGCTCTATAACATCTACGCC | GTGGACCAGAACCTTGTCTAAG |
| TXN2 | TTCAAGACCGAGTGGTCAACA | CACCTCATACTCAATGGCGAG |
| mtHSP70 | CAGTCTTCTGGTGGATTAAGCAAA | CTTCAGCCATATTAACTGCTTCAAC |
| LONP1 | GTTCCCGCGCTTTATCAAGAT | GTAGATTTCATCCAGGCTCTC |
| MCU | TTCCTGGGACATCATGGAGC | TGTCTGTCTCTGGCTTCTGG |
| NCLX | ATGGTGGCTGTGTTCCTGACCT | GGTGCAGAGAATCACAGTGACC |
| LETM1 | CCGAGTGCCTTCGCATAGTG | ACTTCTCTACTACCGAGTCATCG |
| β-actin | TCTTCCAGCCTTCCTTCCTG | CAATGCCTGGGTACATGGTG |
| **Chemicals, enzymes and other reagents** | | |
| DMEM | Life Technologies | 10569010 |
| FBS | Life Technologies | 10270106 |
| Antibiotic-Antimycotic liquid | Life Technologies | 15240062 |
| LIF | Merck Chemicals & Life Science S.A | ESG1107 |
| Retinoic acid | Miltenyi | 130-117-339 |
| Bleomycin | Sigma-Aldrich Merck | B8416 |
| Palbociclib | Absource Diagnostic | S1116 |
| | Ambeed | A295334 |
| Sapanisertib | Absource Diagnostics | S2811 |
| Abemaciclib | MedChem Express | HY-16297 |
| siRNA against CypD | siPOOLs Biotech | |
| Cyclosporin A | Sigma-Aldrich Merck | 30024 |
| NIM811 | MedChemExpress | SDZ NIM811 |
| Ruthenium 360 | Sigma-Aldrich Merck | 557440 |
| CGP37157 | Sigma-Aldrich Merck | C8874 |
| siRNA against NCLX | siPOOLs Biotech | |
| siRNA against MCU | siPOOLs Biotech | |
| BD Horizon™ BV421 Annexin V | Fisher Scientific | 563973 |
| BD Propidium Iodide Staining Solution | Fisher Scientific | 556463 |
| Seahorse medium | Agilent | 103575-100 |
| Seahorse calibrant solution | Agilent | 100840-000 |
| Seahorse cell culture plates and cartridges | Agilent | 102342-100 |
| Glutamine | Life Technologies | 25030024 |
| Glucose | AppliChem | A1422 |
| 4X Laemli buffer | Bio-Rad | 1610747 |
| SDS–PAGE Nu-PAGE 4-12% Bis-Tris gel | Invitrogen | NP0322BOX |
| MES running buffer | Thermofisher Scientific | NP0002 |
| Polyvinylidene difluoride (PVDF) membrane | Immobilon-P | IPFL00010 |
| ECL™ Western Blotting Reagents | Merck Life Science | GERPN2106 |
| Trizol | Invitrogen | 15596018 |
| SYBR-GoTaq BRYT | Promega | A6002 |
| MitoTracker Red CMXRos | Invitrogen | M7512 |
| Rhod-2/AM | Biogen | R1245MP |
| Pluronic® F-127 | Merck Life Science | P2443-250G |
| MitoTracker Green FM | Life Technologies | M7514 |

| Reagent/resource | Reference or source | Identifier or catalog number |
|---|---|---|
| Digitonin | Fisher Scientific | 10636033 |
| Fluo-4-AM | Life Technologies | F14201 |
| MitoSOX | Fisher Scientific | M36008 |
| Image-iT TMRM Reagent | Invitrogen | I34361 |
| BCECF-AM | Invitrogen | B1170 |
| FURA 2-AM | Fisher Scientific | J62728.MCR |
| JC-10 | Stratech | 22204-AA |
| Formalin | Sigma-Aldrich Merck | HT501128-4L |
| Hematoxylin | Roche | 760-2021 |
| Mounting Medium, Toluene-Free | Agilent | CS705 |
| Opti-MEM™ I Reduced Serum Medium | Thermofisher | 31985062 |
| Lipofectamine™ RNAiMAX Transfection Reagent | Thermofisher | 13778075 |
| Adenosine 5′-triphosphate disodium salt hydrate | Merck | A2383-5G |
| Thapsigargin | Invitrogen | T7459 |
| FCCP, mitochondrial oxidative phosphorylation uncoupler | Abcam | ab120081 |
| DC Protein Assay | Bio-Rad | 5000114 |
| Calcein-AM | Fisher Scientific | C3099 |
| Lenti-X Packaging Single Shot | Takara Bio | 631275 |
| **Software** | | |
| Prism | Graphpad | Version 9 |
| FiJi | NIH | |
| Adobe Illustrator | Adobe | 24.1.1 |
| QuPath | | QuPath 0.4. 4 |
| FlowJo | BD Bioscience | |
| **Other** | | |
| X-ray films | Fuji | |
| cover glasses #1,5 25 mm | VWR International Eurolab | MENZCB00250RAC |
| LIVE/DEAD™ Fixable Aqua Dead Cell Stain Kit | Fisher Scientific | L34957 |
| ChromoMap DAB Kit | Roche | 760-159 |
| Blood & Cell Culture Midi kit | Qiagen | 13323 |
| CellTiter-Glo® Luminescent Cell Viability Assay | Promega | G7571 |
| Senescence-associated beta-galactosidase staining kit | Cell Signaling Technology | 9860 |
| iScript™ cDNA Synthesis kit | Bio-Rad | 170-8891 |

## Cell culture and treatments

Embryonic stem cells (ES) cells were grown on a feeder layer of inactivated mouse embryonic fibroblasts (MEFs) with ES medium (EM): Dulbecco's modified Eagle's medium (DMEM, high glucose, Life Technologies) supplemented with 15% heat-inactivated foetal bovine serum (FBS) (Gibco), penicillin/streptomycin (100 U/mL), LIF (1000 U/ml), 0.1 mM non-essential amino acids, 1% glutamax, and 55 mM β-mercaptoethanol. Differentiation was induced by switching into differentiation medium (DM): EM without LIF and supplemented with 0.1 μM retinoic acid (RA). Cells were maintained until passage 15. Senescence was induced by irradiation (5 Gy) or Palbociclib (5 μM, Sellekchem).

A549, IMR90, and SK-MEL-103 cells were maintained in DMEM supplemented with 10% heat-inactivated FBS and Antibiotic-Antimycotic liquid (Life Technologies). Cells were maintained in a humidified incubator at 37 °C with 5% $CO_2$. Senescence was induced by 0.03 U/mL Bleomycin (A549, Sigma), or 1 μM CDK4/6 inhibitor Palbociclib treatment (SK-MEL-103, Absource Diagnostic), or by irradiation (IMR90, 20 Gy). Quiescence was induced by 200 nM Sapanisertib treatment (Absource Diagnostics GmbH). 7 days after the beginning of the senescence- or quiescence-inducing treatment or irradiation, cells were treated with 3 nM siRNA against CypD (siPOOLs Biotech), Cyclosporine A (Sigma), or NIM811 (MedChem-Express), as indicated in the figure legends. siRNA transfection was achieved mixing siPOOLs diluted in Opti-MEM (Gibco) with diluted Lipofectamine RNAiMAX transfection reagent (Invitrogen), with a 1:1 ratio. The mixture was incubated for 5 min at RT and added to previously plated cells, as indicated by the manufacturer. Viability assays were performed after 7 days of genetic depletion of CypD or 6 days of pharmacological inhibition. The one-day difference accounts for the delayed effect of siRNA on protein downregulation compared to the more immediate action of chemical compounds. Functional analyses were performed 5 days after siRNA transfection, before a substantial drop in cellular viability. Calcium levels were pharmacologically modulated by 10 μM Ruthenium 360 or 10 μM CGP-37157 (Sigma-Aldrich Merck) treatments, or by siRNA-mediated downregulation of MCU and NCLX (siPOOLs Biotech), as indicated previously. Culture medium containing pharmacological treatments was replaced every 48 h.

LNCaP cell line (human prostate cancer) was obtained from LGC standards/ATCC, cultivated according to supplier's recommendations, STR profiled, and tested regularly for mycoplasma. Cells were cultured in RPMI medium with 10% foetal bovine serum and 1% penicillin–streptomycin. Cells were maintained in a humidified incubator at 37 °C and 5% $CO_2$. Cellular senescence was induced by treatment with the CDK4/6 inhibitor abemaciclib (250 nM). After 7 days, senescent cells were collected and used for experiments.

## Screening

Pluripotent ES with inducible Cas9 cells were transduced at a low (0.3) multiplicity of infection with a previously described mouse genome-wide sgRNA library (Ruiz et al, 2016). One million cells were induced to differentiate removing Leukemia Inhibition Factor (LIF) and adding 0.1 μM Retinoic acid (RA) for 5 days. Senescence was then induced using 5 μM Palbociclib (Palbo) or 5 Gy irradiation (IR) for 7 days. Then, 2 μg/ml doxycycline (doxy) was added for 10 days to induce Cas9 expression and allow for genomic editing.

## Preparation of gDNA for next-generation sequencing

For the genome-wide screen, genomic DNA was isolated from cell pellets using a genomic DNA isolation kit (Blood & Cell Culture

Midi kit (Qiagen). After gDNA isolation, sgRNAs were amplified and barcoded by PCR as in Shalem et al, 2014, to amplify the DNA fragment containing sgRNA sequences. PCR products were sequenced on a HiSeq 4000 instrument (Illumina) at 50 bp reads to a depth of 30 M reads per sample.

## Analysis of pooled CRISPR screen

To identify negative hits in our screen, reads were pre-processed by removing adapters using cutadapt (Kechin et al, 2017) with parameters "-e 0.2 -a GTTTTAGAGCTAGAAATAGCAAGT-TAAAATA -m 18", and processed to contain only the unique sgRNA sequence. Processed reads were aligned to the sgRNA probe sequences using bowtie (v.0.12.9) (Langmead et al, 2009) with parameters "-n 1 -l 19". The number of reads aligning to each probe and sample were computed in R. The count matrix was normalized using the "varianceStabilizingTransformation" function from the DESeq2 R package (Love et al, 2014). For each gene, a mixed effect linear model was fitted including the sequencing batch as fixed effect and the sgRNA probe as random effect when more than one probe had non-zero counts. Models were fitted with the function "lmer" from the lme4 R package (Bates et al, 2015). Contrasts between groups were performed with the "glht" function from the multcomp R package (Hothorn et al, 2008). Multiple comparison correction was done with the Benjamini-Hochberg algorithm. The samples are: murine Embryonic Stem cells (mES), mES induced to differentiation (Diff) by removing LIF and adding 0.1 μM RA, senescent cells treated with 5 μM Palbociclib and 5 Gy irradiation (Palbo and IR, respectively) and the corresponding condition treated with doxycycline (+ doxy) for the activation of inducible-Cas9 expression. Pairs are (CONDITION [+doxy; after Cas9 activation] vs CONTROL [before activation; sgRNA should be inactive]): Viability (Diff+doxy vs Diff), PD (PD+doxy vs PD), Doxo (Doxo+doxy vs Doxo), IR (IR+doxy vs IR).

## Viability assays

Proliferative, senescent, and quiescent cells were plated in white cell culture 96-well plates and treated as previously indicated. The CellTiter-Glo® Luminescent Cell Viability Assay (Promega) was used as indicated by the manufacturer to determine cell viability of IMR90, A549, and SK-MEL cells at the end of the treatments. LNCaP cells were stained with Annexin V-BV421 and propidium iodide after 6 days of CSA treatment following manufacturer's instructions. Data were acquired with a Becton-Dickinson FACS Celesta flow cytometer and analysed using the FlowJo software. In Appendix Fig. S1, A549 cells were seeded in 12-well cell culture plates and stained with 0.5% crystal violet for 30 min. Following washing with water, the dye was dissolved in methanol and colorimetric intensity was determined at an excitation wavelength of 540 nm using a Microplate Reader.

## Senescence-associated beta-galactosidase staining

Senescence-associated beta-galactosidase staining was carried out using a kit (Cell Signaling Technology) according to manufacturer's instructions.

## Seahorse

Oxygen consumption rate per cell was measured with a Seahorse XFe24 Analyser (Agilent) in intact cells in Seahorse XF medium supplemented with 10 mM glucose, 1 mM pyruvate, and 2 mM glutamine. Seahorse medium, calibrant solution, cell culture plate, and cartridge were purchased and used as indicated by the manufacturer (Agilent Technologies Spain, S.L.). Routine respiration (BASAL) is followed by inhibition of ATP synthase (final concentration 1.5 μM Oligomycin), leading to the non-phosphorylating LEAK state. Subsequently, the maximal capacity of the electron transfer system (MAX) is assessed via two injections of Carbonyl cyanide *m*-chlorophenyl hydrazone (CCCP, 1 μM each). Finally, 1 μM Rotenone and 1 μM Antimycin A completely inhibit respiration.

## SDS–PAGE western blots

Cells were pelleted at 250×*g* at RT and the pellet was washed twice with PBS. Cells were lysed in cold RIPA buffer (25 mM Tris pH 8.0, 150 mM NaCl, 1% Triton-X100, 0.5% Sodium deoxycholate, 0.1% SDS) containing 1× protease inhibitor cocktail (Sigma), 1 mM sodium fluoride, and 1 mM sodium orthovanadate. Protein samples were quantified using a colorimetric DC Protein Assay (detergent compatible, Bio-Rad), following the manufacturer's instructions, and 20 μg of proteins were loaded for each condition. Samples were prepared by mixing the extracted proteins with 4× Laemli buffer (200 mM Tris-HCl pH 6.8, 40% glycerol, 8% w/v Sodium dodecyl sulphate (SDS), 0.04% v/v bromophenol blue, 20% *β*-mercaptoethanol). Samples were resolved by SDS polyacrylamide gel electrophoresis, using SDS–PAGE Nu-PAGE 4-12% Bis-Tris gel (Invitrogen). Gels were run in MES running buffer (50 mM MES, 50 mM Tris Base, 0.1% SDS, 1 mM EDTA, pH 7.3; Thermofisher Scientific). Proteins were transferred to a Polyvinylidene difluoride (PVDF) membrane (Immobilon-P) at 100 V for 1.5 h in Tris-Glycine transfer buffer (25 mM Tris-HCl, 192 mM Glycine), 20% methanol. Membranes were blocked with 10% milk in PBS with 0.1% Tween 20 (PBS-T) for 1 h at RT while shaking and then incubated overnight with specific primary antibodies in 2% milk in PBS-T at 4 °C. Following three washes with PBS-T for 10 min at RT while shaking, membranes were incubated with horseradish peroxidase (HRP)-conjugated anti-mouse or anti-rabbit (Life Technologies) IgG secondary antibodies at a dilution of 1:3000. Blots were developed by incubating membranes with ECL™ Western Blotting Reagents GE Healthcare (Merck Life Science) and visualising the signal on X-ray films (Fuji/Tecnologia, Diagnostico e Investigación, SA) at different exposure times. Films were developed using an X-ray film processor and bands were quantified using Fiji (*Fiji* Is Just ImageJ). Primary antibodies used are indicated in the reagents list.

## RNA extraction and qRT-PCR

In total, $10^6$ cells were harvested by trypsinisation, washed twice in PBS, and collected by centrifugation at 250×*g* at 4 °C for 5 min. RNA was extracted with Trizol (Invitrogen, Waltham, MA, USA) according to the manufacturer's recommendations. Up to 1 μg of total RNA was retro-transcribed into cDNA using an iScript™ cDNA Synthesis kit (Bio-Rad) following the manufacturer's protocol.

Quantitative real-time–PCR was performed using SYBR-GoTaq BRYT (Promega) in a QuantStudio 6 Flex thermocycler (Applied Biosystem). Input normalisation of all the qRT–PCR data was performed by the ΔΔCt method using the housekeeping gene β-actin. Primer sequences used for mRNA analyses are listed in the reagents list.

## Confocal microscopy analysis

Cells were plated on round cover glasses #1, 5 25 mm (VWR International Eurolab, SLU) and imaged live with a Zeiss Elyra PS1 system Image Scanning Microscope (Airyscan) or a Spinning disk confocal microscope with the Andor Revolution system, both equipped with a stage-top incubator and $CO_2$ control system (37 °C, 5% $CO_2$). The excitation and emission wavelengths for each fluorescent dye were selected according to the manufacturer's instructions. Analyses were performed using FiJi.

## Mitochondrial morphology

Cells were incubated with 200 nM MitoTracker Red CMXRos or MitoTracker Green (Invitrogen) in culture DMEM for 15 min at 37 °C. Before imaging, the medium was replaced with fresh DMEM. Mitochondrial volume and length were measured as indicated in (Kakimoto et al, 2021).

## Mitochondrial calcium

Cells were loaded with 2 µM Rhod-2/AM (Biogen) + 0.1% Pluronic® F-127 (Merck Life Science) in imaging buffer (156 mM NaCl, 3 mM KCl, 2 mM $MgSO_4$·$7H_2O$, 1.25 mM $K_2HPO_4$, 2 mM $CaCl_2$, 10 mM HEPES, 10 mM D-glucose, pH 7.4) at RT for 10 min, to allow the dye to localize into mitochondria. MitoTracker Green FM (200 nM, Life Technologies) was added and the cells were moved to 37 °C for 30 min, where the dye is de-esterified and activated. Before analysis, the buffer containing the fluorescent probes was replaced with new imaging buffer and plasma membranes were permeabilised with 10 µM digitonin. For kinetic analyses, cells were plated in black cell culture 96-well plates and loaded with Rhod-2/AM as indicated previously. Experiments were run in a injectors-equipped plate reader (BMG Labtech) and calcium fluxes were stimulated by a 10 µM ATP injection. In all, 20 µM Ruthenium 360 or 20 µM CGP-37157 (Sigma-Aldrich Merck) treatments were added during the incubation time, as indicated in the figure legend.

Imaging of mitochondrial CEPIA-2mt was performed in A549 cells stably expressing the probe. CEPIA-2mt lentiviral plasmid was purchased from VectorBuilder and lentiviral particles were generated in HEK293T packaging cells using the Lenti-X Packaging Single Shots kit (Takara Bio). Twenty-four hours after transduction, cells were selected for puromycin. On the day of the experiment, cells were loaded with MitoTracker Red CMXRos (200 nM, Invitrogen) as indicated previously. Cells were washed in HBSS with 20 mM HEPES buffer (Gibco) and live-imaged.

## Cytosolic calcium

Calcium levels at rest were measured loading cells with 5 µM Fluo-4/AM (Life Technologies) + 0.1% Pluronic® F-127 (Merck Life Science) in Standard Solution (SS: 20 mM HEPES pH 7.4 HBSS and 0.1% bovine serum albumin) for 30 min at 37 °C. Cells were then washed with fresh SS and fluorescence intensity was monitored. To avoid differences due to cellular morphology, 20 z-stacks were taken for each cell.

Cytosolic calcium fluxes were monitored loading cells previously plated in black 96-well plate with 2 µM Fura-2AM in DMEM for 30 min at 37 °C. Cells were washed in HBSS with 20 mM HEPES and loaded in a plate reader equipped with injectors. Fluorescence at 340 nm (ex) and 380 nm (ex) was measured to determine the baseline level for 10 s, followed by injection of 10 µM ATP and subsequent monitoring of Fura-2AM levels for 110 s. Background fluorescence was obtained in wells containing unloaded cells in HBSS buffer and subtracted from the raw fluorescent intensities. Background corrected fluorescence at 340 nm was divided by that at 380 nm to obtain 340/380 ratio.

## ROS

Cells were loaded with 5 µM MitoSOX™ (Fisher Scientific SL) and 200 nM MitoTracker Green FM (Life Technologies) in HBSS/Ca/Mg for 15 min at 37 °C, washed with fresh HBSS buffer, and imaged.

## Visualization of mPTP transient openings by microscopy

mPTP flickering events were visualised as described in (Sambri et al, 2020). Cells were loaded with 100 nM TMRM (Image-iT TMRM Reagent, Invitrogen) and 300 nM Mitotracker Green (Life Technologies) in HBSS buffer for 30 min at 37 °C. Cells were then washed and imaged with an Andor Spinning disk confocal microscope equipped with a stage-top incubator and $CO_2$ control system. Time sequences of 350 images were acquired with a 60× oil-immersion objective (0.05 s exposure for the red channel, 0.075 s exposure for the green channel, 5 z-stacks). The resulting videos were analysed with Fiji. 10 ROIs (10 × 10 µm squares) were randomly generated for each sequence and the fluorescence of the two channels was measured over time. When a flickering event happens, a sudden drop in TMRM fluorescence followed by complete recovery is measured and counted as one event.

## Flow cytometry assay to detect mPTP opening with Calcein-AM

To measure the opening of mPTP by flow cytometry, we used the Mitochondrial Permeability Transition Pore Assay Kit (ab239704) according to manufacturer instructions, with some modifications: briefly, cells were prepared to a final concentration of $10^6$ cells/mL and 1 ml aliquots were distributed into five separate tubes. The MPTP Staining Dye (Calcein-AM) was diluted 1:500 with pre-warmed MPTP Wash Buffer. Tube #1 served as the unstained control. To tubes #2, #3, and #4, we added 5 µl of the Calcein-AM staining. Additionally, 5 µL of $CoCl_2$ were added to tubes #3 and #4, and 5 µl of Ionomycin were added to tube #4. We slightly modified this assay and prepared an additional tube #5 with addition of Calcein-AM, $CoCl_2$ and 20 µM CSA. During the staining phase, tubes were incubated at 37 °C for 15 min protected from light. After this, cells were centrifuged and re-suspended in 1 ml of MPTP Wash Buffer (maintaining CSA in tube #5), this was considered time 0 min (t = 0 min). To measure the rate of mPTP opening, cells were incubated at RT for 30 min and 60 min. After the indicated

incubation times, fluorescence was measured by flow cytometry on a Gallios Beckman Coulter flow cytometer (BD Biosciences).

## Cytosolic pH measurements

Cells were plated in 96-well plates to achieve ~80% confluence on the day of the experiment. Cells were treated with 3 μM fluorescent H+-sensitive dye 2'-7'-bis(carboxyethyl)-5(6)-carboxyfluorescein (BCECF) (Dojin) in an HCO$_3$-containing buffer (110 mM NaCl, 5 mM KCl, 10 mM glucose, 25 mM NaHCO$_3$, 1 mM MgSO$_4$, 1 mM KPO$_4$, and 2 mM CaCl$_2$, pH 7.4) for 20 min, and then washed. Ratios of BCECF fluorescence at Ex/Em = 490/530 were acquired using a Synergy HTX Multimode plate reader. The fluorescence ratios were normalised by protein quantification.

## Membrane potential measurements (ΔΨm) by flow cytometry

Cells were loaded with 100 nM TMRM (Image-iT TMRM Reagent, Invitrogen) and 300 nM Mitotracker Green (Life Technologies) in DMEM for 30 min at 37 °C. Cells were then trypsinised, washed in PBS and stained with 1:1000 in PBS LIVE/DEAD™ Fixable Aqua Dead Cell Stain Kit, for 405 nm excitation (Thermofisher), for 15 min on ice. Two more PBS washes where performed and cells were analysed using flow cytometry, before and after treatment with 10 μM FCCP (Abcam). TMRM signal was normalised to mitochondria mass (Mitotracker Green signal) and background signal was removed (signal after FCCP treatment).

## Membrane potential measurements (ΔΨm) by microscopy

Cells were loaded with 15 μM JC-10 (Stratech) and incubated for 30 min at 37 °C protected from light. Cells were then imaged live with a Zeiss Elyra PS1 system Image Scanning Microscope (Airyscan) as indicated previously. As control, cells were incubated with 20 μM FCCP (Abcam).

## In vivo experiments

For each mouse, SK-MEL-103 cells (one million) were harvested, re-suspended in a volume of 100 μl PBS and injected in the flank of 8 weeks old female athymic nude mice under isofluorane anaesthesia. Tumour volume was measured daily with a caliper and calculated as $V = (a \times b^2)/2$ where $a$ is the longer and $b$ is the shorter of two perpendicular diameters. Treatments were started when the tumour volume was ~100 mm$^3$ and the mice were randomly assigned to control, Palbociclib only, Cyclosporin A only, or Palbociclib + Cyclosporin A treated groups. Palbociclib (Ambeed) was dissolved in 50 mM sodium lactate at 12.5 mg/ml and administered by daily oral gavage at the indicated doses. Cyclosporine A (20 mg/kg/day, Sigma) was dissolved in olive oil and administered via IP injection for 7 days, while controls were injected with the same volume of olive oil. Palbociclib treatment started 2 days before Cyclosporin A administration. After 10 days, the tumours were extracted and formalin fixed for p21 staining. Animal procedures testing senolytic activity on xenografted SK-MEL cells were approved by the Ethics Committee for Animal Experimentation (CEEA) of the Scientific Park of Barcelona (PCB,

license number CEEA-19-029) and the Government of Catalunya and complied with their ethical regulations.

## Immunohistochemical analysis

Samples were fixed overnight at 4 °C with neutral buffered formalin (HT501128-4L, Sigma-Aldrich). Paraffin-embedded tissue sections (2–3 μm) were air dried and further dried at 60 °C overnight. Immunohistochemistry was performed using a Ventana discovery XT. Antigen retrieval for was performed with Cell Conditioning 1 (CC1) buffer starndard conditions (ref: 950-124, Roche). Extra blocking was done with Casein (ref: 760-219, Roche). Staining was performed using the mouse monoclonal (mAb) Anti-Human p21WAF1/Cip1 Clone SX118 (M7202, Dako-Agilent) at 1:50 and incubated for 60 min. Signal was detected using the rabbit mAb to mouse IgG1 + IgG2a + IgG3 [M204-3] (ab133469, Abcam) 1:500 for 30 min. Antigen–antibody complexes were reveled with ChromoMap DAB Kit (ref: 760-159, Roche). Sections were counterstained with hematoxylin (760-2021, Roche) and mounted with Mounting Medium, Toluene-Free (CS705, Agilent) using a Dako CoverStainer.

Specificity of staining was confirmed by staining with a mouse IgG1 [MOPC-21] (Abcam, ab18443) isotype control. Brightfield images were acquired with a NanoZoomer-2.0 HT C9600 digital scanner (Hamamatsu) equipped with a ×20 objective. All images were visualized with a gamma correction set at 1.8 in the image control panel of the NDP.view 2 U12388-01 software (Hamamatsu, Photonics, France). Quantitative analysis was performed using QuPath (Bankhead et al, 2017).

## Statistical analysis

The data were analysed using GraphPad Prism v.9.0.1 software and represented as mean ± SD of at least three independent biological replicates. Statistical analyses between two different groups of data were performed using an unpaired two-tailed Student's *t* test. Means of multiple groups were compared by two-way analysis of variance (ANOVA) with Tukey's correction.

## Materials availability

This study did not generate new unique reagents.

# Data availability

Any additional information required to reanalyse the data reported in this paper is available from the lead contact upon request. Microscopy raw data are available on BioStudies, accession number S-BSST1617.

The source data of this paper are collected in the following database record: biostudies:S-SCDT-10_1038-S44318-024-00259-2.

# Peer review information

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

## Acknowledgements

We are grateful to IRB Microscopy facility and in particular to Dr. Lídia Bardia and Dr. Nikolaos Giakoumakis for the help designing and analysing the imaging experiments, and to Dr. Sébastien Tosi for the generation of the macro used in the analysis of mitochondrial morphology. We would also like to thank the IRB Histopathology and Bioinformatics Units for the technical support. We also thank the Genomics Unit from Scientific and Technological Centers (CCiTUB), Universitat de Barcelona, for the preparation of the library used in the screening. MP was supported by the European Union's Horizon 2021 research and innovation programme under the Marie Sklodowska-Curie grant agreement (HORIZON-MSCA-2021-PF-01) and the Barcelona Institute of Science and Technology (BIST). VLP was recipient of a predoctoral contract from Spanish Ministry of Education (FPU-18/05917). JB, NH, and JM work was funded by the Asociación Española Contra el Cancer (AECC; PRYCO211023SERR) and by the Instituto de Salud Carlos III (CP19/00170). SR was funded by the NIH Intramural Research Program. MK was funded by the Barcelona Institute of Science and technology (BIST) and Asociación Española Contra el Cáncer (AECC; POSTD18020SERR) and supported by the European Molecular Biology Organization (EMBO). Work in the laboratory of MS was funded by the IRB and "laCaixa" Foundation, by a Coordinated-AECC grant (PRYCO211023SERR), and by Secretaria d'Universitats i Recerca del Departament d'Empresa i Coneixement of Catalonia (Grup de Recerca consolidat 2017 SGR 282).

## Author contributions

**Margherita Protasoni**: Conceptualization; Data curation; Formal analysis; Investigation; Methodology; Writing—original draft; Writing—review and editing. **Vanessa López-Polo**: Conceptualization; Investigation; Methodology; Writing—review and editing. **Camille Stephan-Otto Attolini**: Software; Methodology. **Julian Brandariz**: Investigation. **Nicolas Herranz**: Supervision; Methodology. **Joaquin Mateo**: Conceptualization; Supervision; Funding acquisition; Writing—review and editing. **Sergio Ruiz**: Resources; Investigation. **Oscar Fernandez-Capetillo**: Conceptualization; Resources; Writing—review and editing. **Marta Kovatcheva**: Conceptualization; Formal analysis; Supervision; Investigation; Methodology; Writing—review and editing. **Manuel Serrano**: Conceptualization; Supervision; Funding acquisition; Writing—original draft; Writing—review and editing.

Source data underlying figure panels in this paper may have individual authorship assigned. Where available, figure panel/source data authorship is

listed in the following database record: biostudies:S-SCDT-10_1038-S44318-024-00259-2.

## Disclosure and competing interests statement

MK has ongoing or completed research contracts with Galapagos NV, Rejuveron Senescence Therapeutics, and mesoestetic®. MS is shareholder of Senolytic Therapeutics, Inc., Life Biosciences, Inc., Rejuveron Senescence Therapeutics, AG, and Altos Labs, Inc. MS has been consultant, until the end of 2022, of Rejuveron Senescence Therapeutics, AG, and Altos Labs, Inc. The funders had no role in study design, data collection and analysis, decision to publish, or preparation of the manuscript.

# Expanded View Figures

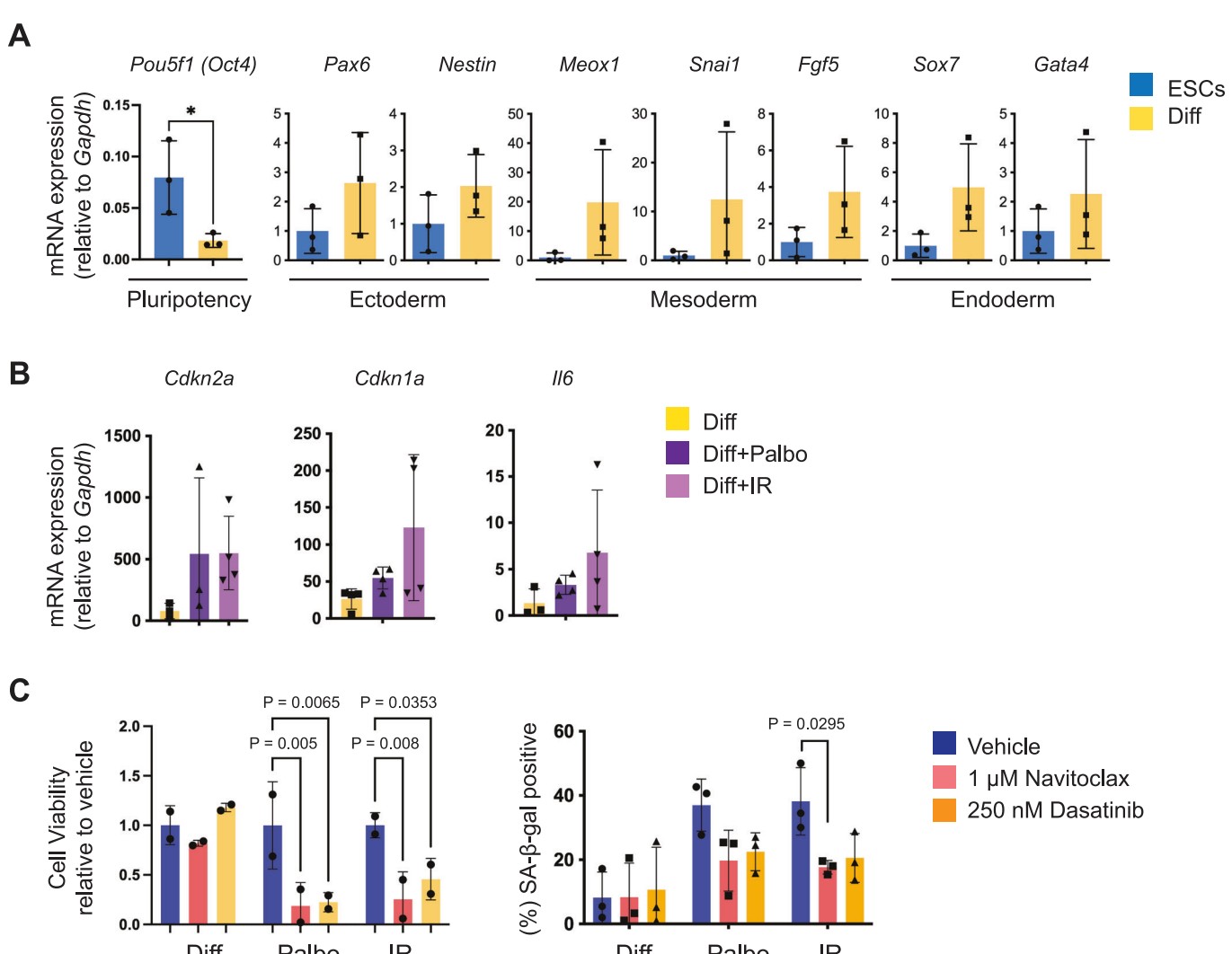

Figure EV1. Validation of the mESC CRISPR/Cas9 screening platform used to identify senolytic targets. Related to Fig. 1.

(A) Relative mRNA expression of markers of ES cells (*Pou5f1/Oct4*) and all three germ layers in cells after differentiation induction (Ectoderm= *Pax6, Nestin*; Mesoderm= *Meox1, Snai1, Fgf5*; Endoderm= *Sox7, Gata4*). Signals were normalized to that of *Gapdh*. *n* = 3 biological replicates. (B) Relative mRNA expression of senescence markers (*Cdkn2a, Cdkn1A, Il6*) in differentiated cells and cells induced to senescence by Palbociclib or irradiation treatment. Signals were normalized to that of GAPDH. *n* = 3-4 biological replicates. (C) Quantification of cell viability and senescence-associated beta-galactosidase (SA-β-gal + ) in differentiated and senescent cells, untreated or after 1 µM navitoclax or 250 nM Dasatinib treatment. *n* = 3 biological replicates. Data shown are mean ± SD. Statistical analyses were performed with 2-way ANOVA multiple comparison with Tukey's correction. *P* values are indicated in the figure.

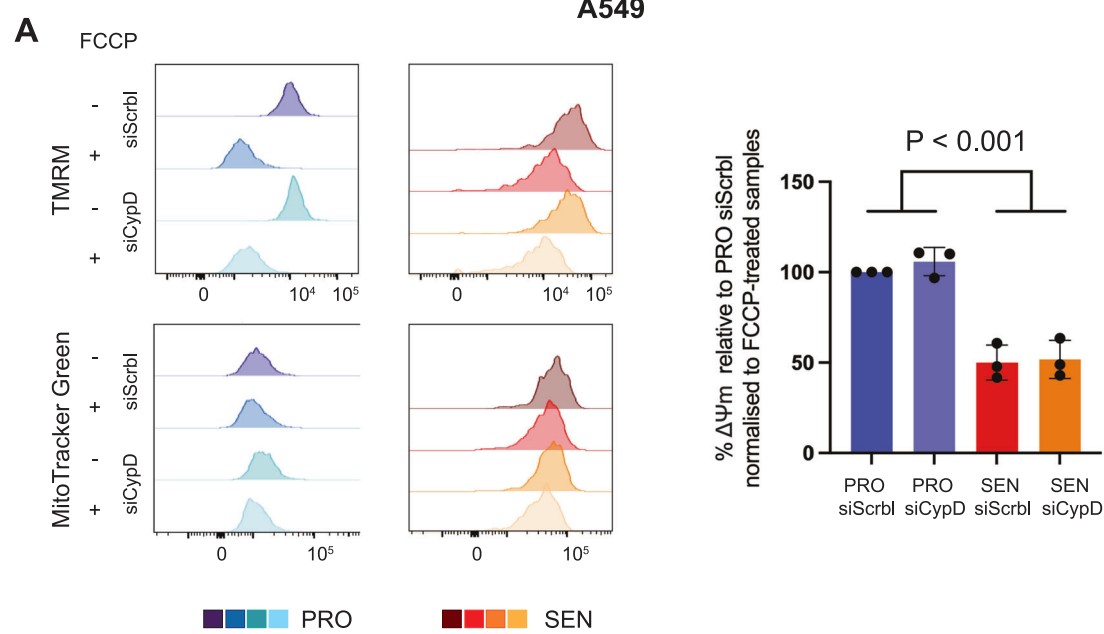

**A549**

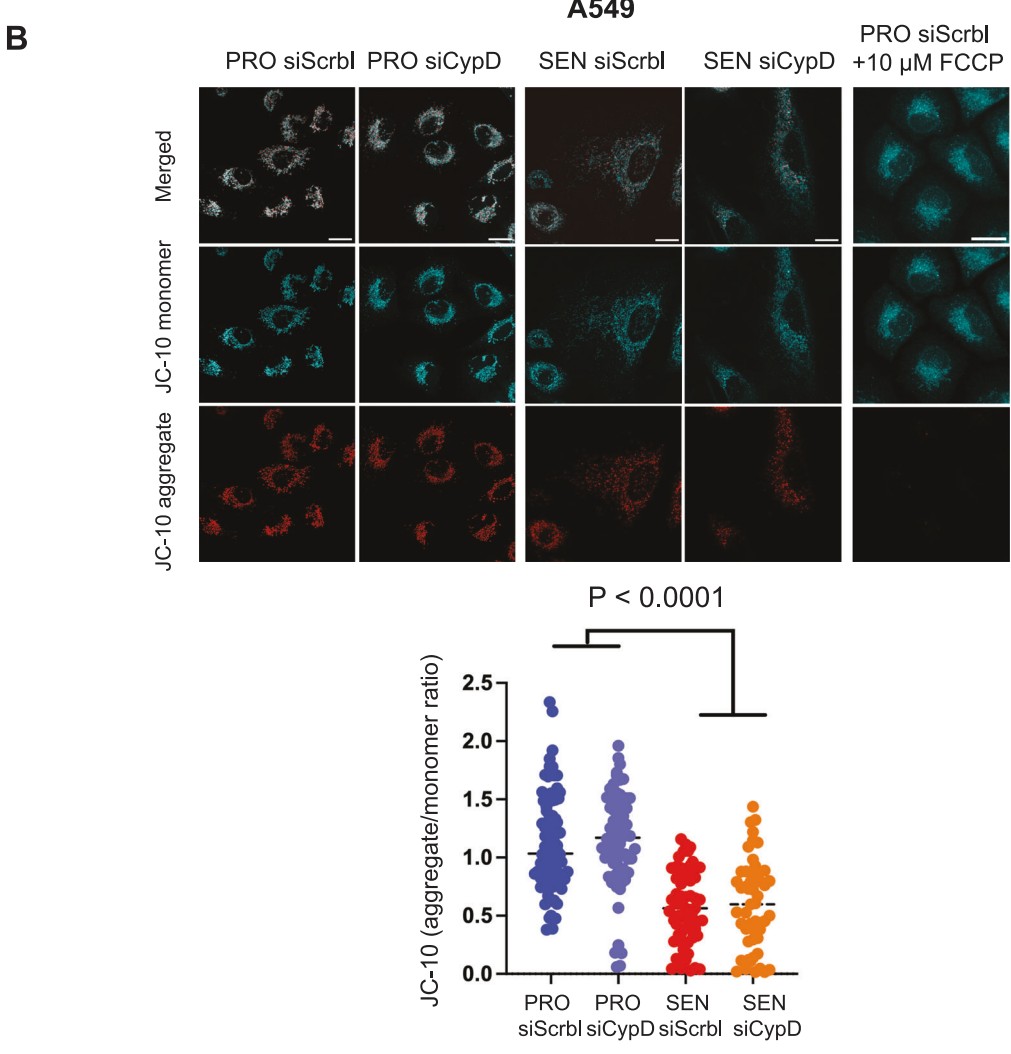

**A549**

◀ **Figure EV2. Mitochondrial membrane potential (ΔΨm) is not affected by cyclophilin D depletion. Related to Figs. 4–6.**

(A) Representative flow cytometry plot and quantifications of proliferating (blue) and senescent (red) A549 cells, treated with siRNA Scrbl or siRNA against CypD for 5 days, as indicated. Senescence was induced with bleomycin. ΔΨm was measured staining the cells with TMRM. Signal was then normalized to mitochondrial mass, measured co-staining the cells with Mitotracker green. Background fluorescence was assessed in the same samples after treatment with FCCP. $n = 3$ independent experiments. (B) Representative images and quantification of JC-10 fluorescence levels in proliferating (blue) senescent (red) A549 cells, 5 days after treatment with siRNA against CypD or siRNA Scrbl. Results are shown as the ratio between red (aggregate) and green (monomer, represented in cyan) signals. $n = 53$–85 cells from a total of 3 independent experiments. The values plotted in the graphs are the mean ± SD. Statistical analyses were performed with 2-way ANOVA multiple comparison with Tukey's correction. *P* values are indicated in the figure.

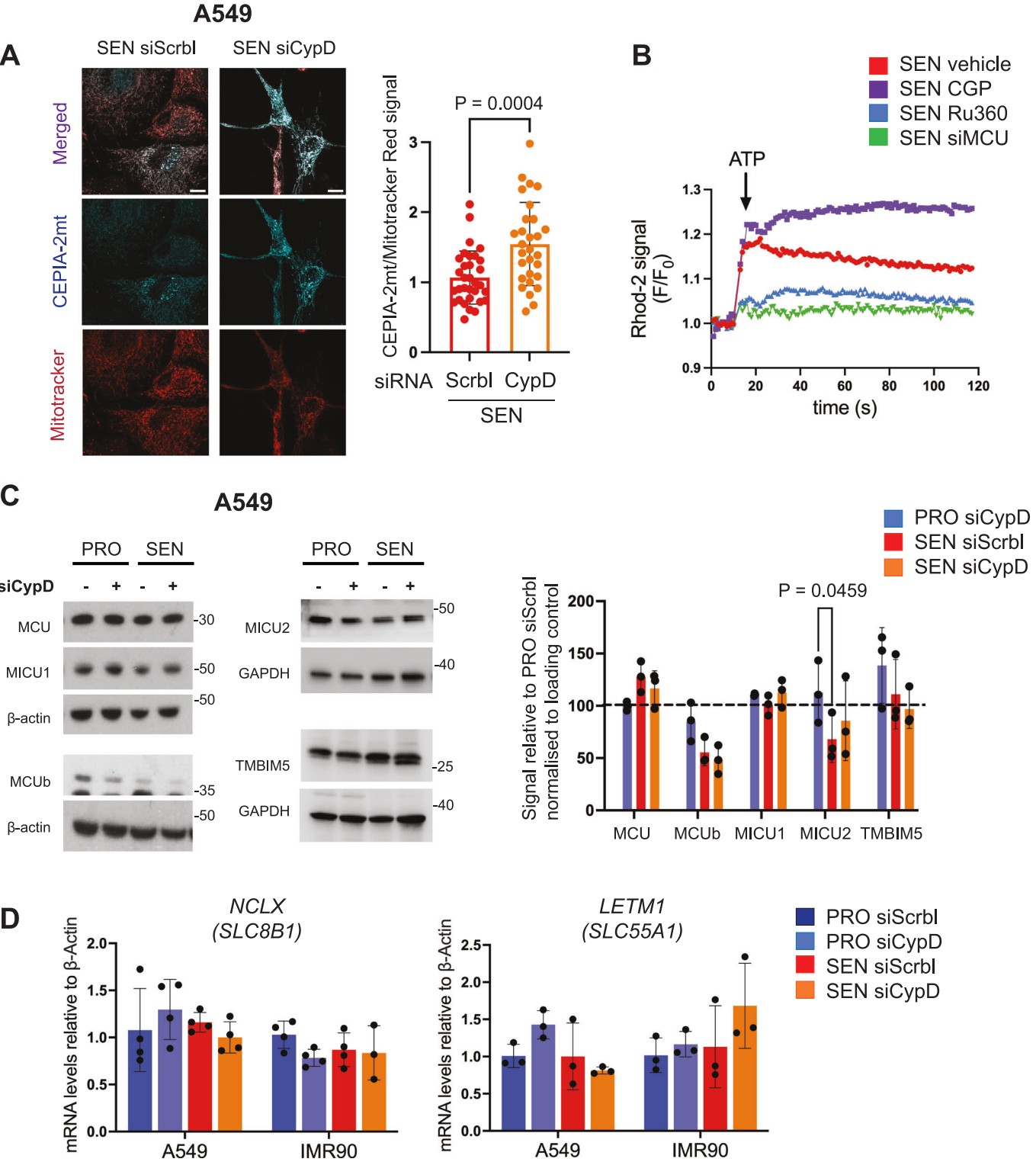

◀ **Figure EV3.  Mitochondrial calcium imbalances are not due to alterations in transporters expression. Related to Fig. 4.**

(**A**) Representative images and quantification of mitochondrial matrix $Ca^{2+}$ levels measured by CEPIA-2mt fluorescent levels in senescent A549 cells 4 days after treatment with siRNA against CypD or siRNA Scrbl. Live-cell images were acquired of cells simultaneously expressing CEPIA-2mt and stained with Mitotracker red. $n = 28–32$ cells from a total of 3 independent experiments. CEPIA-2mt signal was normalized to mitochondrial mass (Mitotracker red signal). (**B**) Average Rhod-2 fluorescent trace in A549 senescent cells treated with vehicle (red trace), pre-treated for 30 min with Ru360 (blue trace), CGP (purple trace), or depleted of MCU (7 days siRNA against MCU, green trace) before and after 10 μM ATP stimulus. $n = 4$ independent experiments. (**C**) Immunodetection of MCU, MICU1, MICU2, MCUb, and TMBIM5 by Western blots of total cell lysates separated by SDS–PAGE. The graphs show the densitometric quantification of the signals corresponding to each protein normalized to that of the loading control, from 3 independent biological replicates of A549 cells. (**D**) Relative mRNA expression of NCLX and LETM1 in proliferating (blue) or senescent (red) A549 and IMR90 cells, treated with siRNA Scrbl or siRNA against CypD for 5 days. Signals were normalized to that of β-actin. $n = 3–4$ biologically independent samples. All the values plotted in the graphs are the mean ± SD. Statistical analyses were performed with 2-way ANOVA multiple comparison with Tukey's correction. $P$ values are indicated in the figure.

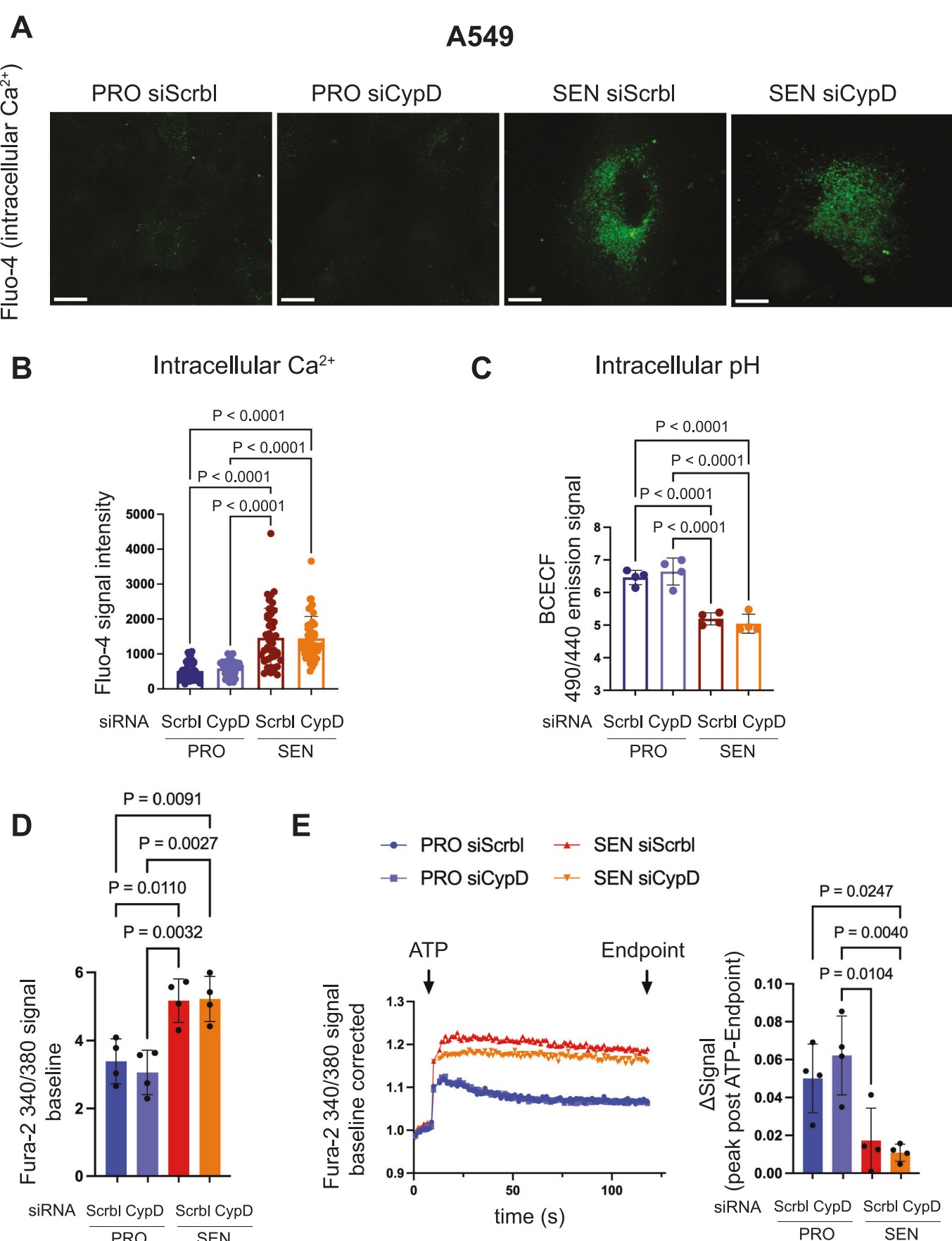

**Figure EV4. Cytosolic calcium and pH are not affected by cyclophilin D depletion. Related to Fig. 4.**

(A) Representative images of cytosolic calcium levels in proliferating and senescent A549 cells, 5 days after treatment with siRNA against CypD or siRNA Scrbl. Live-cell images were acquired of cells stained with Fluo-4 (20 z-stacks). Scale bar = 20 μm. (B) Quantification of Fluo-4 levels from (A). $n = 44$–50 cells from a total of 3 independent experiments. (C) Quantification of BCECF signals in proliferating (blue) and senescent (red) A549 cells, 5 days after treatment with siRNA against CypD or siRNA Scrbl. $n = 4$ independent experiments. (D) Baseline averages of the traces shown in (E). Baseline for each experiment is the average of the first 10 s of measurement, before the ATP injection. $n = 4$ independent experiments. (E) ATP-induced intracellular $Ca^{2+}$ traces (F340/380) and slopes in proliferating (blue) and senescent (red) A549, expressing or depleted of CypD. Cells were loaded with Fura-2 and stimulated with a single injection of 10 μM ATP. Fluorescence signal was recorded every second for 2 min in total. The traces show the average of $n = 4$ independent experiments. Changes in intracellular $Ca^{2+}$ after the stimulus were quantified as the difference between the maximal fluorescence signal measured post ATP injection (peak) and at endpoint (Δ Signal). All the values plotted in the graphs are the mean ± SD. Statistical analyses were performed with 2-way ANOVA multiple comparison with Tukey's correction. *P* values are indicated in the figure.

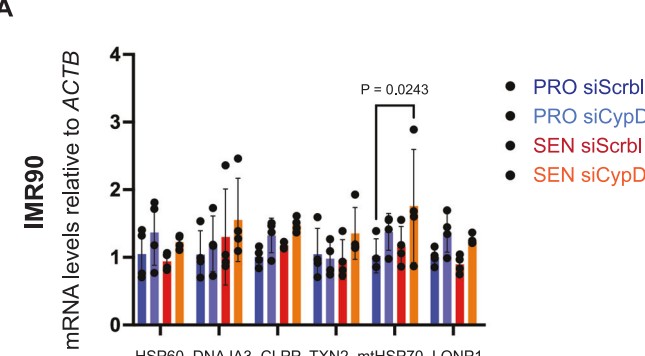

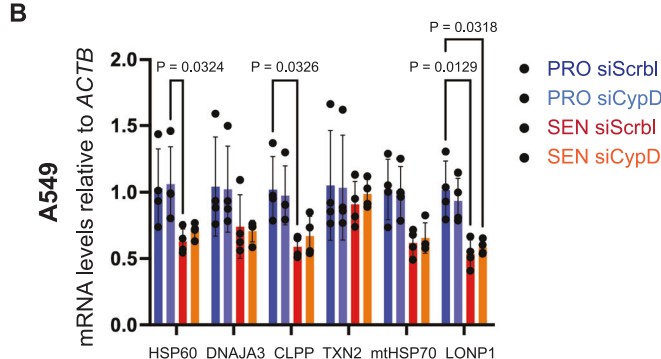

**Figure EV5. Cyclophilin D knockdown does not result in UPRmt-associated transcriptional changes. Related to Fig. 5.**

Relative mRNA expression of UPRmt markers in proliferating (blue) or senescent (red) IMR90 (**A**) and A549 (**B**) cells, treated with siRNA Scrbl or siRNA against CypD for 5 days. Signals were normalized to that of β-actin. n = 3–4 biologically independent samples. All the values plotted in the graphs are the mean ± SD. Statistical analyses were performed with 2-way ANOVA multiple comparison with Tukey's correction. P values are indicated in the figure.

