## [Peer Review File · The EMBO Journal]

Cyclophilin D plays a critical role in the survival of senescent cells

Margherita Protasoni, Vanessa López-Polo, Camille Stephan-Otto Attolini, Julian Brandariz, Nicolas Herranz, Joaquin Mateo, Sergio Ruiz, Oscar Fernandez-Capetillo, Marta Kovatcheva, and Manuel Serrano

Corresponding author(s): Manuel Serrano (mserrano@altoslabs.com), Marta Kovatcheva (marta.kovatcheva@ifom.eu)

Review Timeline:

Submission Date:	20th Oct 23
Editorial Decision:	11th Dec 23
Revision Received:	18th Jul 24
Editorial Decision:	9th Aug 24
Revision Received:	23rd Aug 24
Accepted:	16th Sep 24

Editor: William Teale

Transaction Report:

Dear Manuel,

Thank you for submitting your manuscript for consideration by the EMBO Journal. It has now been seen by three referees whose comments are shown below.

After our discussion on Zoom, I would like to invite you to submit a revised version of the manuscript, addressing the comments of all three reviewers. I should add that it is EMBO Journal policy to allow only a single round of revision, and acceptance of your manuscript will therefore depend on the completeness of your responses in this revised version.

Thank you for the opportunity to consider your work for publication. I look forward to your revision.

Best wishes,

William

William Teale, PhD
Editor
The EMBO Journal
w.teale@embojournal.org

When submitting your revised manuscript, please carefully review the instructions below and include the following items:

- 1) a .docx formatted version of the manuscript text (including legends for main figures, EV figures and tables). Please make sure that the changes are highlighted to be clearly visible.
- 2) individual production quality figure files as .eps, .tif, .jpg (one file per figure).
- 3) a .docx formatted letter INCLUDING the reviewers' reports and your detailed point-by-point response to their comments. As part of the EMBO Press transparent editorial process, the point-by-point response is part of the Review Process File (RPF), which will be published alongside your paper.
- 4) a complete author checklist, which you can download from our author guidelines ([https://wol-prod-cdn.literatumonline.com/pb-assets/embo-site/Author Checklist%20-%20EMBO%20J-1561436015657.xlsx](https://wol-prod-cdn.literatumonline.com/pb-assets/embo-site/Author%20Checklist%20-%20EMBO%20J-1561436015657.xlsx)). Please insert information in the checklist that is also reflected in the manuscript. The completed author checklist will also be part of the RPF.
- 5) Please note that all corresponding authors are required to supply an ORCID ID for their name upon submission of a revised manuscript.
- 6) We require a 'Data Availability' section after the Materials and Methods. Before submitting your revision, primary datasets produced in this study need to be deposited in an appropriate public database, and the accession numbers and database listed under 'Data Availability'. Please remember to provide a reviewer password if the datasets are not yet public (see <https://www.embopress.org/page/journal/14602075/authorguide#datadeposition>). If no data deposition in external databases is needed for this paper, please then state in this section: This study includes no data deposited in external repositories. Note that the Data Availability Section is restricted to new primary data that are part of this study.

Note - All links should resolve to a page where the data can be accessed.

- 7) When assembling figures, please refer to our figure preparation guideline in order to ensure proper formatting and readability in print as well as on screen:
<http://bit.ly/EMBOPressFigurePreparationGuideline>

8) For data quantification: please specify the name of the statistical test used to generate error bars and P values, the number (n) of independent experiments (specify technical or biological replicates) underlying each data point and the test used to calculate p-values in each figure legend. The figure legends should contain a basic description of n, P and the test applied. Graphs must include a description of the bars and the error bars (s.d., s.e.m.).

9) We would also encourage you to include the source data for figure panels that show essential data. Numerical data can be provided as individual .xls or .csv files (including a tab describing the data). For 'blots' or microscopy, uncropped images should be submitted (using a zip archive or a single pdf per main figure if multiple images need to be supplied for one panel). Additional information on source data and instruction on how to label the files are available at .

10) We replaced Supplementary Information with Expanded View (EV) Figures and Tables that are collapsible/expandable online (see examples in <https://www.embopress.org/doi/10.15252/emboj.201695874>). A maximum of 5 EV Figures can be typeset. EV Figures should be cited as 'Figure EV1, Figure EV2' etc. in the text and their respective legends should be included in the main text after the legends of regular figures.

12) Our journal encourages inclusion of *data citations in the reference list* to directly cite datasets that were re-used and obtained from public databases. Data citations in the article text are distinct from normal bibliographical citations and should directly link to the database records from which the data can be accessed. In the main text, data citations are formatted as follows: "Data ref: Smith et al, 2001" or "Data ref: NCBI Sequence Read Archive PRJNA342805, 2017". In the Reference list, data citations must be labeled with "[DATASET]". A data reference must provide the database name, accession number/identifiers and a resolvable link to the landing page from which the data can be accessed at the end of the reference. Further instructions are available at .

Further instructions for preparing your revised manuscript:

- a point-by-point response to the referees' comments, with a detailed description of the changes made (as a word file).
- a word file of the manuscript text.
- individual production quality figure files (one file per figure)
- a complete author checklist, which you can download from our author guidelines (<https://www.embopress.org/page/journal/14602075/authorguide>).

- Expanded View files (replacing Supplementary Information)

We realize that it is difficult to revise to a specific deadline. In the interest of protecting the conceptual advance provided by the work, we recommend a revision within 3 months (10th Mar 2024). Please discuss the revision progress ahead of this time with the editor if you require more time to complete the revisions. Use the link below to submit your revision:

Referee #1:

The paper "A critical role of cyclophilin D in the survival of senescent cells" by Protason et al. provides mechanistic insights into the role of cyclophilin D on cellular senescence. The authors used a CRISPR based screen to identify molecular determinants of senescence. They found that CypD, the only identified mitochondrial protein (from 13 total), might play an important role in senescence. The paper covers important aspects of cell biology relevant for a number of pathological diseases. Nevertheless, I have some comments and suggestions:

Major

To some extent the study is not novel. That mitochondria and calcium are important regulators is well known.

One of the weak points in my opinion is the number of variables in this study. By this I mean: number of cell types, stimuli, duration of experiments etc. I do not understand why the authors decided to use this approach. Among other cells they use two melanoma cells for example. The whole study could have been done with them. This would have made making conclusions much easier.

Some essential experiments have been performed with technologies that are highly prone to artifacts. For example: calcein-based measurements of mitochondrial calcium is not commonly used, TMRE is well established but should be controlled and major findings should be tested with an alternative approach i.e. dye. The same counts for Rhod-2 which is often found not only in the mitochondrial matrix but also in other cellular compartments.

There are number of very reliable genetic sensors for mitochondrial calcium. Why were these not used? This would also allow dynamic measurements of mCa²⁺ and not only end point detection.

Mitox is supposed to measure ROS (includes many oxidants i.e. unspecific) in mitochondria. Is mitox sensitive to changes in the inner membrane potential (flickering)?

Cyclosporin A is a potent calcineurin and thereby NFAT inhibitor. Accordingly, many Calcium-controlled signaling pathways as well as gene expression will be affected upon treatment. Many of these effects will not be directly linked with cellular senescence.

Can reintroduction/overexpression reverse i.e. rescue the effects induced by siRNA against CypD?

Instead of treating the cells with relatively high concentrations of highly unspecific drugs such as RU360 and CGP the authors could have silenced the expression of MCU and/or NCLX. In this regard, the time of treatment namely 5-6 days is way to long and will affect the cellular phenotype not only in a calcium dependent manner.

What is the actual n value in Fig. 3 for example? How many cover-slips and how many transfections are included in the quantification of all events?

Minor

Presentation of microscopy images: signal intensity could be improved (Rhod2 Fig 2B+C, MitoSox 3F, especially cytosolic

calcium Suppl. 3 intensity barely visible).

Scale bars in the microscopy images are missing.

Formatting should be checked e.g. abemaciclib (250nM); Figure legend 1 check font size

Fig 4a. methodology not clear (Calcein-AM plus CoCl₂ measurements by flow cytometry), please briefly explain.

Indicate concentrations of used siRNA, how was knockdown achieved (transfection procedure)?

Please indicate concentration of Ru360 and CGP-37157 in the methods? Was always 10 μM used for 5/6 days as in Fig legend 6C?

Referee #2:

Summary:

Protasoni et al. identify cyclophilin D as a critical player in the survival of senescent cells in vitro that acts by promoting calcium efflux from mitochondria thus preventing calcium overload, preserving mitochondrial morphology, and protecting from caspase 1/4-dependent cell death. The critical role of cyclophilin D in senescence was identified by crispr/cas9 screening of differentiated mouse ES cells treated with x-ray radiation or Palbociclib. It was further validated in human cell lines where cyclophilin D was shown to be upregulated during drug-induced senescence, whereas its silencing or inhibition with CsA led to reduced cell viability. Using fluorescence-based assays the authors show that cyclophilin D protects cells by promoting transient mPTP openings "mitochondrial flickering" and calcium efflux thus preventing calcium overload, alterations in mitochondrial morphology, and ensuing cell death. The key role of mitochondrial calcium in the survival of senescent cells is supported by protective effect of MCU inhibition and increased cell death caused by NCLX inhibition. Thus, the manuscript presents a convincing dataset on the important role of cyclophilin D/mitochondrial calcium signaling in the survival of senescent cells and provide new targets for senolytic drugs for further studies. However, there are issues that can be addressed to improve the quality of the presented data/further research:

Major Concerns/Comments:

1. In vivo data on the CypD role in the survival of senescent cells are limited to one experiment involving CsA treatment. However, this drug has many off-target effects as noted in the discussion by the authors themselves. It might be useful to perform more in vivo tests and include in further experiments another mPTP inhibitor, NIM811, that is believed to lack immunosuppressive activity. Even better would be utilization of the mutant mouse model targeting CypD.
2. A more nuanced or likely interpretation of the identification that CypD is upregulated in Senescent cells could be that the cell is trying to initiate cell death pathways but is unable to complete the signaling or pathway. If this is truly a mitochondrial calcium dependent effect, wouldn't the authors expect an upregulation of more robust calcium efflux proteins such as NCLX? How are the protein levels changing with regards to known mitochondrial calcium exchange proteins? MCU, MCUB, MICU1/2/3, EMRE, NCLX, TMEM165, etc. Western blots should be performed. Also, did these genes show up in the senescent screen?
3. Mitochondrial membrane potential was measured or shown without normalization/calibration to FCCP/CCCP. That confounds the results and also does not make it possible to estimate its value in the resting state. The resting potential would be useful to know as it certainly can be affected by flickering and if so, it can significantly alter mitochondrial calcium dynamics on its own. Furthermore, many fluorescence assays performed in the study heavily rely on potential-dependent dye uptake (rhod-2 for estimation of mitochondrial calcium, mitotracker red for estimation of morphology) and results should be taken with caution if the potential in the cell conditions differ.
4. The utilization of mitochondrial calcium genetic calcium sensors would be a nice confirmation to the data generated using Rhod2. Rhod2 has several issues with specificity.
5. Measuring mitochondrial calcium flux, independent of cytosolic or ER calcium flux is also appropriate. The authors should utilize a permeabilized cell system, with thapsigargin based inhibition of the ER and a ratiometric reporter similar to that utilized by the Muniswamy and Foskett labs.
6. The assay in Fig4A cannot be used to compare mitochondrial calcium levels. It is designed to estimate mPTP openings by CoCl₂ entry to mitochondria and consequent quenching of Calcein fluorescence. Thus, increased calcein fluorescence may reliably indicate the absence of mPTP openings and not the level of calcium in mitochondria. As mentioned above, rhod-2 (potential-dependent) normalized to mitotracker green (potential-independent) is also not the most accurate method to estimate mitochondrial calcium if the mitochondrial potential is different. The most accurate method would be to use a ratiometric mitochondrial reporter.
7. Cytoplasmic calcium levels in FigS3 are estimated by fluorescence intensity of fluo4. It would be more accurate to use a ratiometric dye (e.g., fura-2) or at least to perform calibration to absolute values.
8. Ru360 has limited cell permeability and is unreliable in intact cells. Knockdown or deletion of MCU and NCLX is appropriate for these experiments and will provide more rigor.

9. The first direct data suggesting CypD has a function that is not pro-death and the first direct evidence of it acting as a calcium release valve in intact cells was not cited or referenced (Elrod and Molkentin, JCI, 2010).
10. What's the molecular mechanism for the downregulation of CypD in the context of senescence?

Referee #3:

In this study, the authors conducted a genome-wide CRISPR/Cas9 screening and identified Cyclophilin D (CypD) as a determinant of cellular senescence. The authors further investigated the molecular mechanisms underlying this effect, concluding that in senescent cells, CypD acts as a conduit for the extrusion of mitochondrial Ca²⁺. Consequently, genetic or pharmacological inhibition of CypD leads to senolysis through mitochondrial calcium overload. Overall, the data quality is high, consistent with the standards of the originating laboratory, which is a leader in the field of senescence. However, the proposed mechanism is not entirely convincing due to certain experimental and interpretative limitations that hinder its publication in its present form

- The contribution of the Permeability Transition Pore (PTP) flickering to mitochondrial calcium extrusion remains a subject of intense debate. While some studies have suggested such a contribution (e.g. PMID 20890047), others have explicitly ruled it out (e.g., PMID 24755650). There is a consensus that the role of PTP in mitochondrial calcium efflux is generally limited, except perhaps in specific cell types and/or under certain pathophysiological conditions. Importantly, the authors' own results appear to indicate a marginal contribution of PTP to mitochondrial calcium levels. Specifically, senescent cells exhibit a substantial induction of Cyclophilin D (CypD) (Fig 2A) and its flickering activity (Fig 3), but no discernible difference in baseline calcium levels (Scrbl in Figs 4B-C). Only after 4 days of CypD silencing (why not 2?) is a modest increase in calcium levels observed, which can hardly be defined as "overload."
- I find that the interpretation of the effect of PTP inhibition proposed here is in stark contrast to the existing literature. Inhibition (either genetic or pharmacological) of the PTP has been consistently shown to desensitize PTP opening to mitochondrial calcium overload. This implies that PTP inhibition allows mitochondria to accumulate more calcium (increased calcium retention capacity, CRC), not because an efflux mechanism is inhibited, but because the PTP's ability to sense calcium is inhibited. The consequence is that PTP inhibition has been described as a cytoprotective phenomenon in numerous pathological contexts, whereas here, we observe the diametrically opposite phenomenon. Further supporting this discrepancy, the effects of CGP and CSA on CRC and cell viability are typically distinct, whereas they seem to overlap in this study. This is perplexing as they operate through profoundly different mechanisms. Chronic CGP treatment causes mitochondrial calcium overload and thus PTP opening, while CsA prevent PTP opening even in the presence of mitochondrial calcium overload.
- The authors should compare expression levels of proteins involved in mitochondrial Ca²⁺ transport in normal vs senescent cells, including MCU complex components, NCLX, TMEM165 and LETM1.
- The authors show differences in mitochondrial morphology and ROS (fig 5). It is however not clear if this is cause or consequence of mitochondrial calcium overload.
- The use of Ru360 must be revised. Although widely used in the literature, Ru360 is barely membrane permeant and it is not specific. Its used in intact cell must be avoided (see e.g. PMID 17074387). Please use RNAi for MCU and/or EMRE instead.
- CsA as well is not CypD specific. Nowadays, more specific PTP inhibitors are available (e.g. PMID 26924772)

Point-by-point reply to the Referee's comments
Manuscript EMBOJ-2023-115944

[AUTHORS] We sincerely thank the Reviewers for their time and for their insightful comments which have helped us to improve our manuscript. As detailed below, we have addressed all their points, which has implied the generation and addition of a substantial amount of new data. To our satisfaction, all the new experimental material further supports and reinforces the original messages of our work.

Referee 1

The paper "A critical role of cyclophilin D in the survival of senescent cells" by Protasoni et al. provides mechanistic insights into the role of cyclophilin D on cellular senescence. The authors used a CRISPR based screen to identify molecular determinants of senescence. They found that CypD, the only identified mitochondrial protein (from 13 total), might play an important role in senescence. The paper covers important aspects of cell biology relevant for a number of pathological diseases.

[AUTHORS] We thank the Reviewer for the positive comments about our work.

Nevertheless, I have some comments and suggestions:

Major:

To some extent the study is not novel. That mitochondria and calcium are important regulators is well known.

[AUTHORS] It is true that dysregulation of mitochondrial calcium has been previously reported in cellular senescence, particularly regarding the ER efflux transporter ITPR2 and its negative regulator TRCP3 and their role in overloading mitochondria with calcium¹⁻⁴. Our study advances this understanding by identifying a previously unrecognised mitochondrial feature in senescence. Specifically, the transient opening of the mitochondrial permeability transition pore by Cyclophilin D (CypD) as a safeguard regulator of mitochondrial calcium content. Furthermore, we have demonstrated that this feature of senescent cells can be leveraged to eliminate senescent cells preferentially over non-senescent cells, opening a promising clinical application. We have amended the Introduction to state more clearly the context and novelty of our findings (see lines 60-63).

One of the weeks points in my opinion is the number of variables in this study. By this I mean: number of cell types, stimuli, duration of experiments etc. I do not understand why the authors decided to use this approach. Among other cells they use two melanoma cells for example. The whole study could have been done with them. This would have made making conclusions much easier.

[AUTHORS] We apologize for not having provided a rationale for our choice of cellular models. In short, we primarily use one single cellular model of senescence, namely, A549 human lung carcinoma epithelial cells induced to senescence by bleomycin. Additional models were added to reinforce our findings and assess their conservation across different cell types and different

triggers of senescence. To ensure our observations were not model- or stimulus-specific, we reproduced some of our main results in three additional cell models: human normal fibroblasts (IMR90) induced to senescence by irradiation or doxorubicin; a human melanoma cell line (SK-MEL-103) induced to senescence by a targeted chemotherapeutic agent that inhibits CDK4/6; a human prostate cancer cell line (LNCaP) also treated with a CDK4/6 inhibitor. Results obtained with these different cellular models consistently recapitulated the initial observations in A549, thus strengthening our findings.

Regarding the duration of the experiments: all the experiments, regardless of the cellular model, were initiated 7 days after senescence induction. At that point, cells were treated with siRNAs or chemical inhibitors to measure viability or to characterize mitochondrial biology. In the case of viability experiments, scoring was done 7 days post-siRNA treatment or 6 days after the first administration of the chemical inhibitors (the one-day difference accounts for the delayed effect of siRNA on protein downregulation compared to the more immediate action of chemical compounds). To characterize mitochondrial biology cells were examined at day 5 before a substantial drop in cellular viability.

Based on the comment of the reviewer, we have added clarifications of our experimental rationale in the text (**see lines 535-538**).

Some essential experiments have been performed with technologies that are highly prone to artifacts. For example: calcein-based measurements of mitochondrial calcium is not commonly used, TMRE is well established but should be controlled and major findings should be tested with an alternative approach i.e. dye. The same counts for Rhod-2 which is often found not only in the mitochondrial matrix but also in other cellular compartments. There are number of very reliable genetic sensors for mitochondrial calcium. Why were these not used? This would also allow dynamic measurements of mCa²⁺ and not only end point detection.

[AUTHORS] We thank the reviewer for pointing out these potential caveats, which we addressed below:

Calcein-AM: We agree with the reviewer that Calcein-AM is not a common way to measure Ca²⁺, but to measure opening of the mPTP. We have modified the text accordingly (**see lines 192-199**) and we have moved these data to Figure 3 (where the mPTP is analysed).

TMRM: TMRM has been used in two different kinds of experiments in the final version of the manuscript: to monitor mPTP opening and to measure overall membrane potential. In both cases, a different method has been used to confirm the results obtained with TMRM:

1. Our results with Calcein-AM/CoCl₂ recapitulate the observation that senescent cells rely more on mPTP flickering than proliferative cells (Figure 3).
2. We have measured membrane potential with two alternative methods, TMRM and JC-10. In agreement with previous literature^{5,6}, both methods show that senescent cells have a lower membrane potential compared to proliferative cells (**see new Supplementary Figures 3A-B**). It is important to emphasize that using both methods we do not observe changes in mitochondrial potential when CypD is downregulated. Therefore, measurements performed in senescent cells with or without CypD

downregulation using positively charged probes (eg Rhod-2) should not reflect differences in mitochondrial potential.

Supplementary Figure 3

Rhod-2: We have taken several precautions to be sure that our observations are specific for mitochondrial matrix Ca^{2+} and are not artefacts due to the mis-localisation of the probe.

Firstly, in the experiments shown in Figures 4 and 6, we reduced potential cytoplasmic signal by permeabilising cells with 20 $\mu\text{g}/\text{ml}$ digitonin, and we verified the correct localisation of the probe co-staining the cells with Mitotracker Green, which is highly specific for mitochondria. In

our images, we can observe high level of co-localisation between Rhod-2 and Mitotracker green.

To further validate the accurate localisation of the probe, we calculated the co-occurrence of the signals from the two channels (Rhod-2 and Mitotracker Green) by measuring the Mander's Overlap Coefficient (MOV) of 20 ROI sections (20x20 μm) from images collected for the experiment shown in Figure 4B (20 proliferative and 20 senescent siScrbl cells). This coefficient was chosen because it quantifies the extent to which two signals overlap independently of their changes in intensity. The high level of co-occurrence between the two signals confirms clear co-localisation between Rhod-2 and Mitotracker Green. This analysis was performed in Fiji using the Colocalisation Finder macro.

The kinetic analysis of Ca^{2+} is a great suggestion by the reviewer. We have monitored Rhod-2 fluorescence over time using a plate reader in A549 senescent cells, both before and after the injection of 10 μM ATP. As a control, we compared control cells, cells pre-treated for half an hour with 20 μM CGP or 20 μM Ru360, or cells depleted of MCU via siRNA treatment. As expected, Ca^{2+} uptake was extremely reduced in cells where MCU was inhibited by Ru360 or downregulated while the calcium extrusion rate decreased in cells treated with CGP (see **new Supplementary Figure 4B**). These results reinforce the concept that MCU is the main influx transporter of Ca^{2+} and that NCLX has a relevant contribution in Ca^{2+} efflux in senescent cells, in addition to confirming that the detected Rhod-2 signal is specific for mitochondria.

Supplementary Figure 4B

We then monitored Ca^{2+} fluxes in A549 senescent cells treated with siRNA against CypD or Scrbl. Reproducing our previous data, baseline levels were higher in cells treated with siRNA against CypD for 4 days compared to control. Notably, the ATP stimulus induced a significantly higher influx of Ca^{2+} into the mitochondria of senescent cells expressing CypD, while downregulation of the protein reduced the organelles' ability to buffer Ca^{2+} (see new Figure 4D). This observation supports our initial results and suggests that the increased Ca^{2+} steady-state level in these cells hinders their capacity to efficiently regulate Ca^{2+} fluxes.

Figure 4D

CEPIA-2mt: As suggested by the reviewer, we have used a genetic sensor for mitochondrial matrix calcium. Specifically, we have generated a stable A549 cell line expressing a lentiviral-encoded Ca^{2+} probe CEPIA-2mt ⁷. We then induced senescence with bleomycin, and we imaged with a confocal microscope senescent A549-CEPIA-2mt cells transfected or not with siCypD. We quantified the fluorescent signal of CEPIA-2mt normalised to mitochondrial mass (Mitotracker Red signal) in 28 to 32 cells from 3 different coverslips using Fiji. In agreement with the Rhod-2 data, Ca^{2+} levels measured with CEPIA-2mt were increased in senescent cells with downregulated CypD compared to untreated senescent cells (see new Supplementary Figure 4A).

Supplementary Figure 4A

Mitoxox is supposed to measure ROS (includes many oxidants i.e. unspecific) in mitochondria. Is mitoxox sensitive to changes in the inner membrane potential (flickering)?

[AUTHORS] We thank the reviewer for this comment. The reviewer correctly indicates that MitoSOX, being positively charged, is sensitive to mitochondrial membrane potential. As discussed in a previous reply, senescent cells, compared to their non-senescent counterparts, are known to present lower mitochondrial potential^{5,6} and elevated levels of MitoSOX signal⁸. Prompted by the comment of reviewer, we have now compared the levels of membrane potential in senescent cells treated or not with siCypD (see new Supplementary Figure 3A,B), observing no changes in TMRM/MitoGreen ratio. This complements the data indicating no changes in MitoSOX signal in senescent cells after CypD depletion. Taking into account that flickering is a rapid transient process (lasting a few seconds) that involves a fraction of mitochondria per cell (please, see for example Figure 3A of our manuscript), we interpret that blocking flickering does not have a noticeable effect on basal mitochondrial potential or overall MitoSOX signal. We have amended the text to include this concept (see lines 213-219).

Cyclosporin A is a potent calcineurin and thereby NFAT inhibitor. Accordingly, many Calcium-controlled signaling pathways as well as gene expression will be affected upon treatment. Many of these effects will not be directly linked with cellular senescence.

[AUTHORS] We agree that CSA has non-CypD specific effects that should be considered when interpreting the results. For this reason, most of our data in vitro are collected from cells genetically downregulating CypD and our experiments using CSA only confirm and complement these results. To further strengthen our data, we repeated some of our viability assays using NIM811, a CSA-derivative that binds and inhibits cyclophilins, but is unable to form a ternary complex with NFAT and therefore lacks immunosuppressive activity⁹. Interestingly, NIM811 presented similar or even more potent senolytic activity (selective loss of viability of senescent cells) than CSA in A549 and IMR90 cells (see new Figure 2E).

Figure 2E

Can reintroduction/overexpression reverse i.e. rescue the effects induced by siRNA against CypD?

[AUTHORS] We would like to make the reviewer aware of the difficulties of doing rescue experiments with siPOOLS (commercialized by the company siTOOLS). Each siPOOL is a combination of 30 individual siRNAs, each at a very low concentration, all targeting the gene of interest. This design is unique to siTOOLS and is done to minimise the chance of off-target effects: the effective concentration of each siRNA in the pool is so low, that each on its own would not be able to induce knockdown of any gene (whether on- or off-target); it is only through the combined on-target effect of all 30 siRNAs together that target knockdown is achieved:

(<https://www.sitoolsbiotech.com/products/sipools/sipools-product-information>).

Having the above in consideration, a siRNA-resistant CypD construct should be mutated in at least 30 sites to completely desensitize the CypD-encoding cDNA from downregulation by the pool of 30 siRNAs. We believe that in the light of all the other experiments, we provide a compelling argument to involve CypD-release of mitochondrial Ca^{2+} as a key survival mechanism of senescent cells. This other evidence includes chemical inhibition of CypD by CSA and by the CSA-derivative NIM811, and also rescue of senescent cell death by targeting MCU genetically and chemically.

Instead of treating the cells with relatively high concentrations of highly unspecific drugs such as RU360 and CGP the authors could have silenced the expression of MCU and/or NCLX. In this regard, the time of treatment namely 5-6 days is way to long and will affect the cellular phenotype not only in a calcium dependent manner.

[AUTHORS] We thank the reviewer for the comment, and we agree with the suggestion. Consequently, we repeated both the viability and mitochondrial morphology experiments after treating proliferating and senescent cells with siRNAs (siPOOLS) targeting MCU and NCLX (see new Figure 6G-L). The results we obtained closely mirror those from the original experiments with pharmacological inhibitors, thereby reinforcing our model.

Figure 6G-L

We agree that long treatments may trigger complex responses beyond their primary effect. However, at the same time, we aim to recapitulate the effect of siRNAs that usually require several days to produce a full effect on protein levels and eventually impact on cell survival. Indeed, we found that our chronic treatments did not impact non-senescent cells in their survival or in mitochondrial morphology (see previous Figures 6E,F); and treatment of senescent cells with chemical inhibitors nicely reproduce the effects of siRNAs.

What is the actual n value in Fig. 3 for example? How many cover-slips and how many transfections are included in the quantification of all events?

[AUTHORS] We apologize for not explaining this. We have carefully revised all the legends to clearly state information about the n values. In the case of Figure 3, we imaged n=12-13 individual cells per condition, and these individual cells were coming from a total of 3-5 biological independent samples of proliferative cultures or senescent cultures. These 3-5 independent samples were transfected and treated independently, in different days. No more than 3 videos were taken from the same coverslip to reduce cellular and mitochondrial stress.

Minor.

Presentation of microscopy images: signal intensity could be improved (Rhod2 Fig 2B+C, MitoSox 3F, especially cytosolic calcium Suppl. 3 intensity barely visible). Scale bars in the microscopy images are missing.

[AUTHORS] Thanks for mentioning this. We have improved the quality of all the microscopy images by equally increasing brightness in all the pictures from the same experiment. Scale bars have been added where missing.

Formatting should be checked e.g. abemaciclib (250nM); Figure legend 1 check font size

[AUTHORS] We have carefully revised formatting across the entire manuscript. Thank you.

Fig 4a. methodology not clear (Calcein-AM plus CoCl₂ measurements by flow cytometry), please briefly explain.

[AUTHORS] We apologize for not explaining the protocol clearly. We have added a detailed explanation.

Indicate concentrations of used siRNA, how was knockdown achieved (transfection procedure)?

[AUTHORS] We apologise for the oversight. The following text has now been added to the Methods section to better explain how the knockdown was achieved:

“...cells were treated with 3 nM siRNA against CypD (siPOOLS Biotech)... siRNA transfection was achieved mixing siPOOLS diluted in OptiMEM (Gibco) with diluted Lipofectamine RNAiMAX transfection reagent (Invitrogen), with a 1:1 ratio. The mixture was incubated for 5 minutes at RT and added to previously plated cells, as indicated by the manufacturer.”

Please indicate concentration of Ru360 and CGP-37157 in the methods? Was always 10 μM used for 5/6 days as in Fig legend 6C?

[AUTHORS] Cells were treated with 10 μM Ru360 or 10 μM CGP-37157, every 48 hours for 5 or 6 days. The final concentrations are now stated clearly in the Methods section.

Referee 2

Summary:

Protasoni et al. identify cyclophilin D as a critical player in the survival of senescent cells in vitro that acts by promoting calcium efflux from mitochondria thus preventing calcium overload, preserving mitochondrial morphology, and protecting from caspase 1/4-dependent cell death. The critical role of cyclophilin D in senescence was identified by crispr/cas9 screening of differentiated mouse ES cells treated with x-ray radiation or Palbociclib. It was further validated in human cell lines where cyclophilin D was shown to be upregulated during drug-induced senescence, whereas its silencing or inhibition with CsA led to reduced cell viability. Using fluorescence-based assays the authors show that cyclophilin D protects cells by promoting transient mPTP openings "mitochondrial flickering" and calcium efflux thus preventing calcium overload, alterations in mitochondrial morphology, and ensuing cell death. The key role of mitochondrial calcium in the survival of senescent cells is supported by protective effect of MCU inhibition and increased cell death caused by NCLX inhibition. Thus, the manuscript presents a convincing dataset on the important role of cyclophilin D/mitochondrial calcium signaling in the survival of senescent cells and provide new targets for senolytic drugs for further studies.

[AUTHORS] We thank the Reviewer for the positive comments about our work.

However, there are issues that can be addressed to improve the quality of the presented data/further research:

Major Concerns/Comments:

1. In vivo data on the CypD role in the survival of senescent cells are limited to one experiment involving CsA treatment. However, this drug has many off-target effects as noted in the discussion by the authors themselves. It might be useful to perform more in vivo tests and include in further experiments another mPTP inhibitor, NIM811, that is believed to lack immunosuppressive activity. Even better would be utilization of the mutant mouse model targeting CypD.

[AUTHORS] The suggestion to use NIM811 is extremely valuable. We have evaluated the effect of NIM811 in senescent A549 and IMR90 cells. In both models, NIM811 exhibited similar or even more potent senolytic activity (selective loss of viability of senescent cells) (see new Figure 2E).

Figure 2E

Regarding the *in vivo* data, our experiments were conducted in immunodeficient mice, which should not be impacted by the immunosuppressive activity of CSA. Moreover, CSA was only

effective on tumors with high levels of senescence (due to pre-treatment with Palbociclib), but it was ineffective on tumors treated with vehicle. While we agree that additional animal experimentation might be useful, we hope that the current level of evidence together with ethical considerations make them unnecessary.

2. A more nuanced or likely interpretation of the identification that CypD is upregulated in Senescent cells could be that the cell is trying to initiate cell death pathways but is unable to complete the signaling or pathway. If this is truly a mitochondrial calcium dependent effect, wouldn't the authors expect an upregulation of more robust calcium efflux proteins such as NCLX? How are the protein levels changing with regards to known mitochondrial calcium exchange proteins? MCU, MCUB, MICU1/2/3, EMRE, NCLX, TMBIM5, etc. Western blots should be performed. Also, did these genes show up in the senescent screen?

[AUTHORS] We found the reviewer's interpretation very interesting and, as recommended, we performed Western blots analysis of several MCU complex subunits (MCU, MICU1, MICU2 and MCUB) and TMBIM5 in proliferative and senescent A549, with or without siCypD. Interestingly, we observed that senescent cells have reduced levels of MCUB and a modest decrease in MICU2 levels (see new Supplementary Figure 4C). The levels of the other mentioned proteins were similar between senescent and proliferative cells. Considering that MCUB is a dominant-negative subunit of MCU, we speculate that its reduced levels may contribute to the dysregulated Ca^{2+} influx into the mitochondria of senescent cells (see lines 251-255).

The antibodies we tested against NCLX and LETM1 did not work or appeared to be unspecific. To evaluate the mRNA expression levels of NCLX and LETM1, therefore, we performed qPCR analysis in both A549 and IMR90. None of these mRNAs was significantly changed in senescent cells compared to proliferative cells (see new Supplementary Figure 4D).

None of the genes encoding these proteins was identified as important for the survival of senescent cells in the genetic screen.

Supplementary Figure 4C,D

3. Mitochondrial membrane potential was measured or shown without normalization/calibration to FCCP/CCCP. That confounds the results and also does not make it possible to estimate its value in the resting state. The resting potential would be useful to know as it certainly can be affected by flickering and if so, it can significantly alter mitochondrial calcium dynamics on its own. Furthermore, many fluorescence assays performed in the study heavily rely on potential-dependent dye uptake (rhod-2 for estimation of mitochondrial calcium, mitotracker red for estimation of morphology) and results should be taken with caution if the potential in the cell conditions differ.

[AUTHORS] We thank the reviewer for the comment. In the original manuscript, we measured membrane potential solely as a qualitative indicator of mPTP opening. We agree that this does not accurately assess overall membrane potential at the resting state. To address this, we have conducted additional experiments (Fig. S3) where we measured membrane potential normalized to mitochondrial mass and FCCP treatment. Consistent with previous literature^{5,6}, senescent cells present lower membrane potential compared to proliferative cells (**see new Supplementary Figure 3A,B**). As suggested by the reviewer, we have revised the text to be cautious when comparing proliferative and senescent cells using potential-dependent dyes.

We did not observe significant differences in membrane potential when we downregulated CypD in senescent cells (and neither in proliferative cells) (**see new Supplementary Figure 3A,B**). In this regard, our main conclusions derived from the comparison between senescent cells expressing or not CypD remain unaffected.

Supplementary Figure 3

4. The utilization of mitochondrial calcium genetic calcium sensors would be a nice confirmation to the data generated using Rhod2. Rhod2 has several issues with specificity.

[AUTHORS] We would like to kindly refer the reviewer to a similar suggestion posed by Referee 1 (see pages 3-5). In short, we have used the calcium genetic reporter CEPIA-2mt and confirmed that downregulation of CypD results in a significant increase in mitochondrial Ca^{2+} levels (see new Supplementary Figure 4A).

Supplementary Figure 4A

Regarding Rhod2 specificity, we have performed all our measurements using mild digitonin permeabilization to reduce potential cytosolic signal^{10,11}, and we have quantified colocalization with Mitotracker (see data in page 4).

5. Measuring mitochondrial calcium flux, independent of cytosolic or ER calcium flux is also appropriate. The authors should utilize a permeabilized cell system, with thapsigargin based inhibition of the ER and a ratiometric reporter similar to that utilized by the Muniswamy and Foskett labs.

[AUTHORS] We appreciate the reviewer's suggestion to analyse Ca^{2+} fluxes. Unfortunately, we don't have the required instrument to replicate the measurements indicated by the reviewer; and we have been unsuccessful when trying to adapt it to a fluorescent plate reader. As an alternative, we have monitored Rhod-2 fluorescence kinetics using a plate reader in A549 senescent cells, before and after stimulation with 10 μ M ATP. Firstly, we monitored Rhod-2 fluorescence over time in cells pre-treated for half an hour with 20 μ M CGP or 20 μ M Ru360, or depleted of MCU via siRNA treatment. As expected, Ca^{2+} uptake was extremely reduced in cells where MCU was inhibited by Ru360 or downregulated while the Ca^{2+} extrusion rate decreased in cells treated with CGP (see **new Supplementary Figure 4B**). These results indicate that the Rhod-2 signal that we are monitoring is reflecting mitochondrial Ca^{2+} levels. Also, they reinforce the concept that MCU is the main influx transporter of Ca^{2+} and that NCLX has a relevant contribution in Ca^{2+} efflux in senescent cells. We then monitored Ca^{2+} fluxes in A549 senescent cells treated with siRNA against CypD or Scrbl. Reproducing our previous data, baseline levels were higher in cells treated with siRNA against CypD for 4 days compared to control. Notably, the ATP stimulus induced a significantly higher influx of Ca^{2+} into the mitochondria of senescent cells expressing CypD, while downregulation of the protein reduced the organelles' ability to buffer Ca^{2+} (see **new Figure 4D**). This observation supports our initial results and suggests

Supplementary Figure 4B

that the increased Ca^{2+} steady-state level in these cells hinders their capacity to efficiently regulate Ca^{2+} fluxes.

Figure 4D

6. The assay in Fig4A cannot be used to compare mitochondrial calcium levels. It is designed to estimate mPTP openings by CoCl₂ entry to mitochondria and consequent quenching of Calcein fluorescence. Thus, increased calcein fluorescence may reliably indicate the absence of mPTP openings and not the level of calcium in mitochondria. As mentioned above, rhod-2 (potential-dependent) normalized to mitotracker green (potential-independent) is also not the most accurate method to estimate mitochondrial calcium if the mitochondrial potential is different. The most accurate method would be to use a ratiometric mitochondrial reporter.

[AUTHORS] We agree with the reviewer that Calcein-AM is not a common way to measure Ca^{2+} but to measure opening of the mPTP. We have modified the text accordingly (**see lines 192-199**) and we have moved these data to Figure 3 (where the mPTP is analysed).

Regarding the impact of mitochondrial membrane potential of senescent cells vs. proliferative cells, as we have discussed in a previous reply, we have measured it using two alternative methods, TMRM and JC-10. Both methods show that senescent cells have a lower membrane potential compared to proliferative cells (**see new Supplementary Figures 3A,B**). This finding indicates that directly comparing proliferative and senescent cells should consider this confounding factor. However, our conclusions are primarily based on the differences between cells expressing and downregulating CypD. In this regard, it is important to emphasize that we do not observe changes in mitochondrial potential when CypD is downregulated. Therefore, the measurements performed in senescent cells with or without CypD downregulation should not reflect differences in mitochondrial potential.

7. Cytoplasmic calcium levels in FigS3 are estimated by fluorescence intensity of fluo4. It would be more accurate to use a ratiometric dye (e.g., fura-2) or at least to perform calibration to absolute values.

[AUTHORS] As suggested, we performed kinetic analysis of intracellular Ca^{2+} using Fura-2 in a injectors-equipped plate reader (**see new Supplementary Figures 5D,E**). In the first 10 seconds, Fura-2 signal was measured at baseline showing similar results to what we observed with Fluo-4, *i.e.* increased intracellular Ca^{2+} in senescent cells, but no evident differences after CypD depletion. We then measured fluorescence after ATP stimulus. The Ca^{2+} fluxes did not

change significantly between cells treated with siScrbl and siCypD, but interestingly, senescent cells showed a higher peak after stimulus and maintain higher Ca^{2+} levels for longer.

Supplementary Figure 5D,E

8. Ru360 has limited cell permeability and is unreliable in intact cells. Knockdown or deletion of MCU and NCLX is appropriate for these experiments and will provide more rigor.

[AUTHORS] We thank the reviewer for the comment and agree with the suggestion. As a result, we performed both viability and mitochondrial morphology experiments after treating A549 and IMR90 proliferating and senescent cells with siRNA targeting MCU and NCLX (see new Figures 6G-L). The results were very similar to those from the original experiments using pharmacological inhibitors, thereby reinforcing our conclusions.

Figure 6G-L

9. The first direct data suggesting CypD has a function that is not pro-death and the first direct evidence of it acting as a calcium release valve in intact cells was not cited or referenced (Elrod and Molkentin, JCI, 2010).

[AUTHORS] We apologize for this oversight. The reference has now been added to the manuscript.

10. What's the molecular mechanism for the downregulation of CypD in the context of senescence?

[AUTHORS] The protein levels of CypD are modestly elevated in senescent cells (~2 fold) compared to proliferative cells (data shown in Figure 2A), while the mRNA levels are not changed in senescent cells (data shown in Supplementary Figure 2A). The regulation of CypD levels by post-translational mechanisms is a topic essentially unexplored, but certainly important for our future studies.

Referee 3

In this study, the authors conducted a genome-wide CRISPR/Cas9 screening and identified Cyclophilin D (CypD) as a determinant of cellular senescence. The authors further investigated the molecular mechanisms underlying this effect, concluding that in senescent cells, CypD acts as a conduit for the extrusion of mitochondrial Ca^{2+} . Consequently, genetic or pharmacological inhibition of CypD leads to senolysis through mitochondrial calcium overload. Overall, the data quality is high, consistent with the standards of the originating laboratory, which is a leader in the field of senescence.

[AUTHORS] We thank the Reviewer for the positive comments about our work.

However, the proposed mechanism is not entirely convincing due to certain experimental and interpretative limitations that hinder its publication in its present form

- The contribution of the Permeability Transition Pore (PTP) flickering to mitochondrial calcium extrusion remains a subject of intense debate. While some studies have suggested such a contribution (e.g. PMID 20890047), others have explicitly ruled it out (e.g., PMID 24755650). There is a consensus that the role of PTP in mitochondrial calcium efflux is generally limited, except perhaps in specific cell types and/or under certain pathophysiological conditions.

[AUTHORS] We thank the reviewer for raising this important discussion. We agree that the role of the mPTP, due to its complex functionality, continues to be a topic of debate. As the reviewer points out, the relevance of mPTP flickering may be restricted to specific cellular states. In this regard, we do not see mitochondrial alterations in normal cells when CypD is inhibited, suggesting that mPTP flickering is not an important homeostatic process under normal cellular conditions. Senescent cells are known to have major mitochondrial alterations and dysfunctions, although little is known about the exact molecular players. Our work uncovers CypD/mPTP flickering as a relevant process in senescent cells for the maintenance of mitochondrial Ca^{2+} homeostasis. Accordingly, inhibition of this pathway results in loss of viability selectively in senescent cells.

Importantly, the authors' own results appear to indicate a marginal contribution of PTP to mitochondrial calcium levels. Specifically, senescent cells exhibit a substantial induction of Cyclophilin D (CypD) (Fig 2A) and its flickering activity (Fig 3), but no discernible difference in baseline calcium levels (Scrbl in Figs 4B-C). Only after 4 days of CypD silencing (why not 2?) is a modest increase in calcium levels observed, which can hardly be defined as "overload."

[AUTHORS] We understand the point of the reviewer about the similar basal levels of mitochondrial Ca^{2+} when comparing proliferative and senescent cells. There are two aspects that we consider important:

- Senescent cells have a lower mitochondrial potential, as previously reported ^{5,6,12,13} and as shown by us now in the revised version of the manuscript using two different probes (TMRM and JC-10) (see new Supplementary Figure 3A,B). Considering that Rhod-2 is sensitive to mitochondrial potential, we may underestimate the actual baseline levels of Ca^{2+} in senescent cells compared to proliferative cells.
- Regarding Ca^{2+} accumulation in senescent cells upon downregulation of CypD, we acknowledge that the term 'overload' is misleading, and we have replaced it by Ca^{2+}

accumulation. We observe an increase that oscillates between 40 and 55% of the normal state. Although this increase might not seem extensive, it is likely sustained over time and capable of inducing cellular stress. For reference, the magnitude of Ca^{2+} increase that we observe by inhibiting CypD in senescent cells is comparable to what has been reported in other studies manipulating well-established Ca^{2+} regulators (e.g. NCLX downregulation in ¹⁴, or MCU overexpression in ¹⁵. This suggests that the increase we measured is realistic.

- Regarding the timing, it is important to bear in mind that siRNAs take some days to achieve maximal reduction in protein levels.

Supplementary Figure 3

- I find that the interpretation of the effect of PTP inhibition proposed here is in stark contrast to the existing literature. Inhibition (either genetic or pharmacological) of the PTP has been consistently shown to desensitize PTP opening to mitochondrial calcium overload. This implies that PTP inhibition allows mitochondria to accumulate more calcium (increased calcium retention capacity, CRC), not because an efflux mechanism is inhibited, but because the PTP's ability to sense calcium is inhibited. The consequence is that PTP inhibition has

been described as a cytoprotective phenomenon in numerous pathological contexts, whereas here, we observe the diametrically opposite phenomenon.

[AUTHORS] We thank the reviewer for the opportunity to further discuss our data and interpretation. The inhibition of mPTP opening indeed has cytoprotective effects in several models of acute damage, such as cardiac and cerebral ischemia/reperfusion¹⁶⁻¹⁸. In these contexts of acute damage, mPTP inhibition is beneficial. Senescent cells can be described as cells adapted to chronic damage and, therefore, we find conceivable that certain processes may operate differently from cells upon acute damage. Our data present such a novel scenario in which sustained mPTP inhibition is detrimental for the survival of senescent cells, while it has no detrimental effects on normal cells. We have added a few sentences in the Discussion to highlight this important distinction (**see lines 399-402**).

Further supporting this discrepancy, the effects of CGP and CSA on CRC and cell viability are typically distinct, whereas they seem to overlap in this study. This is perplexing as they operate through profoundly different mechanisms. Chronic CGP treatment causes mitochondrial calcium overload and thus PTP opening, while CsA prevent PTP opening even in the presence of mitochondrial calcium overload.

[AUTHORS] We acknowledge that in other models, using proliferative cells, chronic CGP treatment can trigger mPTP opening as a safeguard mechanism to avoid Ca²⁺ accumulation. Based on our observations with senescent cells, we interpret that the influx of Ca²⁺ via MCU (as previously reported in^{1,2}) requires both Ca²⁺ efflux mechanisms, NCLX and mPTP, to maintain homeostatic Ca²⁺ levels. We discuss this fundamental difference in behaviour between proliferative and senescent cells in the Discussion (**see lines 430-432**).

- The authors should compare expression levels of proteins involved in mitochondrial Ca²⁺ transport in normal vs senescent cells, including MCU complex components, NCLX, TMBIM5 and LETM1.

[AUTHORS] We agree with the reviewer's recommendation, and we performed Western blots analysis of several MCU complex subunits (MCU, MICU1, MICU2 and MCUB) and TMBIM5 in proliferative and senescent A549, expressing or depleted of CypD. Interestingly, we observed that senescent cells have reduced levels of MCUB and a modest decrease in MICU2 levels (**see new Supplementary Figure 4C**). The levels of the other mentioned proteins were similar between senescent and proliferative cells. Considering that MCUB is a dominant-negative subunit of MCU, we speculate that its reduced levels may contribute to the dysregulated Ca²⁺ influx into the mitochondria of senescent cells (**see lines 251-255**). The levels of all these proteins were not affected by the downregulation of CypD, suggesting that the effects of CypD inhibition on Ca²⁺ levels are not confounded by alterations in the levels of these proteins.

In the case of NCLX and LETM1, none of the tested antibodies produced reliable detection. Nonetheless, we considered it essential to assess the expression levels of these two transporters, so we measured NCLX and LETM1 mRNA expression levels via qPCR analysis in both A549 and IMR90 cells. None of these mRNAs was significantly changed in senescent cells compared to proliferative cells (**see new Supplementary Figure 4D**).

Supplementary Figure 4C,D

- The authors show differences in mitochondrial morphology and ROS (fig 5). It is however not clear if this is cause or consequence of mitochondrial calcium overload.

[AUTHORS] This is an interesting point. Senescent cells are known to have an elongated and branched mitochondrial network^{19,20} and higher levels of mitochondrial ROS⁸; and we have confirmed both key features in our work (see Figures 5A-D and 5F). Interestingly, upon mitochondrial Ca²⁺ accumulation in senescent cells (by siCypD, or siNCLX, or CGP inhibition of NCLX) we observed fragmentation of the mitochondrial network (see Figures 5A-D, and 6E,F,I,L). Therefore, we interpret that fragmentation is a consequence of Ca²⁺ overload. These observations are in agreement with previous studies showing that mitochondrial Ca²⁺ overload leads to fragmentation^{21,22}. In the case of mitochondrial ROS, we do not observe changes when cells are treated with siCypD (see Figure 5F).

- The use of Ru360 must be revised. Although widely used in the literature, Ru360 is barely membrane permeant and it is not specific. Its used in intact cell must be avoided (see e.g. PMID 17074387). Please use RNAi for MCU and/or EMRE instead.

[AUTHORS] We thank the reviewer for the comment and agree with the suggestion. As a result, we decided to repeat both viability and mitochondrial morphology experiments downregulating both MCU and NCLX with siRNA in proliferating and senescent A549 and IMR90 cells (see new Figures 6G-L). The results were very similar to those from the original experiments using pharmacological inhibitors, thereby reinforcing our model.

Figure 6G-L

• CsA as well is not CypD specific. Nowadays, more specific PTP inhibitors are available (e.g. PMID 26924772)

[AUTHORS] We agree that CSA has non-CypD specific effects that should be considered when interpreting the results. For this reason, most of our *in vitro* data are collected from cells genetically downregulating CypD and our experiments using CSA confirm and complement these results. Nonetheless, as suggested by the reviewer, we have repeated the viability assays in A549 and IMR90 proliferative and senescent cells using NIM811, a cyclophilin inhibitor without immunosuppressive activity (described in the review indicated by the reviewer) (see new Figure 2E). NIM811 showed similar or even more potent senolytic effect (preferential loss of survival of senescent cells) than CSA in both cell models.

Figure 2E

References

1. Ahumada-Castro, U., Puebla-Huerta, A., Cuevas-Espinoza, V., Lovy, A., and Cardenas, J.C. (2021). Keeping zombies alive: The ER-mitochondria Ca²⁺ transfer in cellular senescence. *Biochim Biophys Acta Mol Cell Res* 1868, 119099. <https://doi.org/10.1016/j.bbamcr.2021.119099>.
2. Martin, N., Zhu, K., Czarnecka-Herok, J., Vernier, M., and Bernard, D. (2023). Regulation and role of calcium in cellular senescence. *Cell Calcium* 110, 102701. <https://doi.org/10.1016/j.ceca.2023.102701>.
3. Wiel, C., Lallet-Daher, H., Gitenay, D., Gras, B., Le Calvé, B., Augert, A., Ferrand, M., Prevarskaya, N., Simonnet, H., Vindrieux, D., et al. (2014). Endoplasmic reticulum calcium release through ITPR2 channels leads to mitochondrial calcium accumulation and senescence. *Nat Commun* 5, 3792. <https://doi.org/10.1038/ncomms4792>.
4. Farfariello, V., Gordienko, D.V., Mesilmany, L., Touil, Y., Germain, E., Fliniaux, I., Desruelles, E., Gkika, D., Roudbaraki, M., Shapovalov, G., et al. (2022). TRPC3 shapes the ER-mitochondria Ca²⁺ transfer characterizing tumour-promoting senescence. *Nat Commun* 13, 956. <https://doi.org/10.1038/s41467-022-28597-x>.
5. Passos, J.F., Saretzki, G., Ahmed, S., Nelson, G., Richter, T., Peters, H., Wappler, I., Birket, M.J., Harold, G., Schaeuble, K., et al. (2007). Mitochondrial dysfunction accounts for the stochastic heterogeneity in telomere-dependent senescence. *PLoS Biol* 5, e110. <https://doi.org/10.1371/journal.pbio.0050110>.
6. Hutter, E., Renner, K., Pfister, G., Stöckl, P., Jansen-Dürr, P., and Gnaiger, E. (2004). Senescence-associated changes in respiration and oxidative phosphorylation in primary human fibroblasts. *Biochem J* 380, 919–928. <https://doi.org/10.1042/BJ20040095>.
7. Suzuki, J., Kanemaru, K., Ishii, K., Ohkura, M., Okubo, Y., and Iino, M. (2014). Imaging intraorganellar Ca²⁺ at subcellular resolution using CEPIA. *Nat Commun* 5, 4153. <https://doi.org/10.1038/ncomms5153>.
8. Correia-Melo, C., Marques, F.D.M., Anderson, R., Hewitt, G., Hewitt, R., Cole, J., Carroll, B.M., Miwa, S., Birch, J., Merz, A., et al. (2016). Mitochondria are required for pro-ageing features of the senescent phenotype. *EMBO J* 35, 724–742. <https://doi.org/10.15252/embj.201592862>.
9. Zulian, A., Rizzo, E., Schiavone, M., Palma, E., Tagliavini, F., Blaauw, B., Merlini, L., Maraldi, N.M., Sabatelli, P., Braghetta, P., et al. (2014). NIM811, a cyclophilin inhibitor without immunosuppressive activity, is beneficial in collagen VI congenital muscular dystrophy models. *Hum Mol Genet* 23, 5353–5363. <https://doi.org/10.1093/hmg/ddu254>.
10. Pacher, P., Sharma, K., Csordás, G., Zhu, Y., and Hajnóczy, G. (2008). Uncoupling of ER-mitochondrial calcium communication by transforming growth factor- β . *Am J Physiol Renal Physiol* 295, F1303–F1312. <https://doi.org/10.1152/ajprenal.90343.2008>.
11. Davidson, S.M., and Duchon, M.R. (2018). Imaging Mitochondrial Calcium Fluxes with Fluorescent Probes and Single- or Two-Photon Confocal Microscopy. *Methods Mol Biol* 1782, 171–186. https://doi.org/10.1007/978-1-4939-7831-1_10.
12. Passos, J.F., Nelson, G., Wang, C., Richter, T., Simillion, C., Proctor, C.J., Miwa, S., Olijslagers, S., Hallinan, J., Wipat, A., et al. (2010). Feedback between p21 and reactive

- oxygen production is necessary for cell senescence. *Mol Syst Biol* 6, 347. <https://doi.org/10.1038/msb.2010.5>.
13. Moiseeva, O., Bourdeau, V., Roux, A., Deschênes-Simard, X., and Ferbeyre, G. (2009). Mitochondrial dysfunction contributes to oncogene-induced senescence. *Mol Cell Biol* 29, 4495–4507. <https://doi.org/10.1128/MCB.01868-08>.
 14. Stavsky, A., Stoler, O., Kostic, M., Katoshevsky, T., Assali, E.A., Savic, I., Amitai, Y., Prokisch, H., Leiz, S., Daumer-Haas, C., et al. (2021). Aberrant activity of mitochondrial NCLX is linked to impaired synaptic transmission and is associated with mental retardation. *Commun Biol* 4, 1–14. <https://doi.org/10.1038/s42003-021-02114-0>.
 15. Qiu, J., Tan, Y.-W., Hagenston, A.M., Martel, M.-A., Kneisel, N., Skehel, P.A., Wyllie, D.J.A., Bading, H., and Hardingham, G.E. (2013). Mitochondrial calcium uniporter Mcu controls excitotoxicity and is transcriptionally repressed by neuroprotective nuclear calcium signals. *Nat Commun* 4, 2034. <https://doi.org/10.1038/ncomms3034>.
 16. Matsumoto, S., Friberg, H., Ferrand-Drake, M., and Wieloch, T. (1999). Blockade of the Mitochondrial Permeability Transition Pore Diminishes Infarct Size in the Rat after Transient Middle Cerebral Artery Occlusion. *J Cereb Blood Flow Metab* 19, 736–741. <https://doi.org/10.1097/00004647-199907000-00002>.
 17. Zhang, C.-X., Cheng, Y., Liu, D.-Z., Liu, M., Cui, H., Zhang, B., Mei, Q.-B., and Zhou, S.-Y. (2019). Mitochondria-targeted cyclosporin A delivery system to treat myocardial ischemia reperfusion injury of rats. *J Nanobiotechnology* 17, 18. <https://doi.org/10.1186/s12951-019-0451-9>.
 18. Schinzel, A.C., Takeuchi, O., Huang, Z., Fisher, J.K., Zhou, Z., Rubens, J., Hetz, C., Danial, N.N., Moskowitz, M.A., and Korsmeyer, S.J. (2005). Cyclophilin D is a component of mitochondrial permeability transition and mediates neuronal cell death after focal cerebral ischemia. *Proc Natl Acad Sci U S A* 102, 12005–12010. <https://doi.org/10.1073/pnas.0505294102>.
 19. Vasileiou, P.V.S., Evangelou, K., Vlasis, K., Fildisis, G., Panayiotidis, M.I., Chronopoulos, E., Passias, P.-G., Kouloukoussa, M., Gorgoulis, V.G., and Havaki, S. (2019). Mitochondrial Homeostasis and Cellular Senescence. *Cells* 8, 686. <https://doi.org/10.3390/cells8070686>.
 20. Tak, H., Cha, S., Hong, Y., Jung, M., Ryu, S., Han, S., Jeong, S.M., Kim, W., and Lee, E.K. (2024). The miR-30-5p/TIA-1 axis directs cellular senescence by regulating mitochondrial dynamics. *Cell Death Dis* 15, 1–11. <https://doi.org/10.1038/s41419-024-06797-1>.
 21. Hom, J.R., Gewandter, J.S., Michael, L., Sheu, S.-S., and Yoon, Y. (2007). Thapsigargin induces biphasic fragmentation of mitochondria through calcium-mediated mitochondrial fission and apoptosis. *Journal of Cellular Physiology* 212, 498–508. <https://doi.org/10.1002/jcp.21051>.
 22. Chakrabarti, R., Ji, W.-K., Stan, R.V., de Juan Sanz, J., Ryan, T.A., and Higgs, H.N. (2017). INF2-mediated actin polymerization at the ER stimulates mitochondrial calcium uptake, inner membrane constriction, and division. *Journal of Cell Biology* 217, 251–268. <https://doi.org/10.1083/jcb.201709111>.

Dear Manuel,

Thank you for submitting your revised manuscript. It was sent to the three original referees. Comments have now been returned by one of them. In order not to slow the process down too much, I have decided to move forward towards publication with only this one report. This progress is contingent on no substantial technical concerns being raised by either of the other two referees (should they submit reports). As you will see, though, Referee #3 is in favour of publication.

Before I write to you with a list of minor editorial changes that need to be made to your manuscript, I need to ask you to submit the Source Data associated with the manuscript (image and data files). This is because some potential duplications have been detected in Figure 6. It is far preferable to sort out any potential issues at the earliest possible stage. Please email me if you need any further clarification on what we need or the best way to get it to us.

Yours sincerely,

William

William Teale, PhD
Editor
The EMBO Journal
w.teale@embojournal.org

We realize that it is difficult to revise to a specific deadline. In the interest of protecting the conceptual advance provided by the work, we recommend a revision within 3 months (7th Nov 2024). Please discuss the revision progress ahead of this time with the editor if you require more time to complete the revisions. Use the link below to submit your revision:

Referee #3:

The authors successfully addressed most of my original critiques.

I think there is now an error in supplementary figures 5D,E. S5D shows higher basal $[Ca^{2+}]$ in SEN cells. Baseline corrected traces (I guess calculated as R/R_0) in fig S5E also show higher ATP-induced Ca^{2+} responses in SEN cells (red/orange lines). However, quantification of the "slope" is lower in SEN cells. Maybe colors were inverted in the middle panel, or calculation of the slope is wrong. My suggestion is to simply show traces of 340/380 ratio, since it is genuinely proportional to $[Ca^{2+}]$, with no need to correct for baseline.

All editorial and formatting issues were resolved by the authors.

Dear Manuel,

I am pleased to inform you that your manuscript has been accepted for publication in the EMBO Journal.

Congratulations on a really thought-provoking piece of work!

Best wishes,

William

William Teale, PhD
Editor
The EMBO Journal
w.teale@embojournal.org
